# RAF inhibitors activate the integrated stress response by direct activation of GCN2

Rebecca Gilley[1] ✉, Andrew M. Kidger [1,7], Graham Neill[2], Eve Morrison[1],
Paul Severson [3,8], Dominic P. Byrne[4], Niall S. Kenneth [4], Gideon Bollag[3,9],
Chao Zhang[3,10], Taiana Maia de Oliveira [5], Patrick A. Eyers [4],
Richard Bayliss [6], Glenn R. Masson [2] & Simon J. Cook [1] ✉

Paradoxical activation of wild type RAF by chemical RAF inhibitors (RAFi) is a well-understood 'on-target' biological and clinical response. In this study, we show that a range of RAFi drive ERK1/2-independent activation of the Unfolded Protein Response (UPR), including expression of ATF4 and CHOP, that requires the translation initiation factor eIF2α. RAFi-induced ATF4 and CHOP expression was not reversed by inhibition of PERK, a known upstream activator of the eIF2α-dependent Integrated Stress Response (ISR). Rather, RAFi exposure activated GCN2, an alternate eIF2α kinase, leading to eIF2α-dependent (and ERK1/2-independent) ATF4 and CHOP expression. The GCN2 kinase inhibitor A-92, GCN2 RNAi, GCN2 knock-out or ISRIB (an eIF2α antagonist) all reversed RAFi-induced expression of ATF4 and CHOP indicating that RAFi require GCN2 to activate the ISR. RAFi also activated full-length recombinant GCN2 in vitro and in cells, generating a characteristic 'bell-shaped' concentration-response curve, reminiscent of RAFi-driven paradoxical activation of WT RAF dimers. Activation of the ISR by RAFi was abolished by a GCN2 kinase dead mutation. A M802A GCN2 gatekeeper mutant was activated at lower RAFi concentrations, demonstrating that RAFi bind directly to the GCN2 kinase domain; this is supported by mechanistic structural models of RAFi interaction with GCN2. Since the ISR is a critical pathway for determining cell survival or death, our observations may be relevant to the clinical use of RAFi, where paradoxical GCN2 activation is a previously unappreciated off-target effect that may modulate tumour cell responses.

The RAS-regulated RAF-MEK1/2-ERK1/2 signalling pathway is activated in a variety of cancers due to mutations in RAS (especially KRAS), BRAF and more rarely CRAF, MEK1 or MEK2[1–4]. The most common BRAF mutations, BRAF[V600E/K], are found in melanoma, hairy cell leukaemia, thyroid cancer and colorectal cancer and can signal as constitutively active monomers to drive MEK1/2-ERK1/2 activation and disease progression[5,6]. BRAF[V600E/K] mutants are effectively inhibited by the clinically approved RAF inhibitors (RAFi) Vemurafenib, Dabrafenib and Encorafenib[3,7–9], which are now used in combination with MEK1/2 inhibitors to treat BRAF[V600E/K] mutant metastatic melanoma[10] and colorectal cancer[11].

In contrast, a variety of RAFi have consistently failed to progress through the clinic for tumours with wild type (WT) RAF, including those with RAS mutations, because they cause paradoxical activation of RAF and ERK1/2 signalling[12,13]. This is because WT RAF proteins signal as RAS-dependent homo- or heterodimers in which one protomer can transactivate the other. At sub-saturating doses RAFi promote dimerization[14–16] and whilst binding of RAFi to one protomer inhibits it,

this drives allosteric transactivation of the inhibitor-free, ATP-bound dimer partner, resulting in drug-induced activation of MEK1/2-ERK1/2 in cells. Furthermore, RAFi binding to one protomer of the dimer reduces the binding affinity and drug occupancy time of the second protomer[17]. This negative cooperativity means that only at high doses of RAFi are both dimer partners inhibited and ERK1/2 activation blocked; consequently, RAFi elicit a characteristic broad, bell-shaped concentration-response curve for ERK1/2 activation in cells containing WT RAF. This paradoxical ERK1/2 activation limits efficacy in BRAF wild type tumours but also drives adventitious growth of low-grade tumours in tissue that lacks BRAF[V600E/K] [18]. RAFi-induced activation of ERK1/2 can be mitigated by combination with a MEKi such as Trametinib; this combination enhances ERK1/2 inhibition in BRAF[V600E/K] tumour cells but antagonises paradoxical ERK1/2 activation in tissue that lacks BRAF mutations, thereby providing a large, BRAF[V600E/K] tumour-selective therapeutic window[3,4]. Attempts to overcome paradoxical RAF activation in RAS mutant cancer cells have included the development pan-RAFi that inhibit all RAF isoforms (ARAF, BRAF and CRAF)[19] and 'Paradox Breaker' RAFis[20]. However, whilst these inhibitors show the desired effects on ERK1/2 activity in cells, they have not yet proved successful in clinical trials.

Whilst studying paradoxical RAF activation, we observed that many RAFi promoted a rapid inhibition of DNA replication. Hyperactivation of ERK1/2 signalling beyond a critical 'sweet spot' can drive cell cycle arrest, senescence and even cell death[21–23]; however, we found that the RAFi-induced inhibition of proliferation was independent of MEK1/2-ERK1/2 activation.

Here we show that multiple RAFi, including all three clinically approved drugs (Vemurafenib, Dabrafenib and Encorafenib), pan-RAFi and Paradox Breaking RAFi induce rapid ERK1/2-independent activation of the Integrated Stress Response (ISR)[24] to inhibit DNA synthesis. Mechanistically, RAFi drive ISR activation and cell cycle arrest by binding directly to GCN2 dimers and activating them in a manner reminiscent of the paradoxical activation of wild type RAF dimers by RAFi[13,18]. Since the ISR is a critical pathway that determines cell survival or cell death after stress[25], these observations may be relevant to the further clinical optimisation of RAFi in the context of cancer cell stress responses.

## Results

### RAF inhibitors drive a rapid, ERK1/2-independent inhibition of DNA synthesis

We analysed p-ERK1/2 and EdU incorporation using high-content microscopy[26,27] in NCI-H358 cells (KRAS[G12C]-mutant lung adenocarcinoma cells) treated with Dabrafenib, a clinically approved RAFi (Fig. 1A). Dabrafenib stimulated paradoxical activation of RAF, seen by the characteristic, broad bell-shaped concentration-response curve for p-ERK1/2. Given the adventitious tumour growth observed in the clinic[13,18], we were surprised to see that Dabrafenib actually inhibited DNA synthesis, monitored by EdU incorporation (Fig. 1A). The decline in EdU incorporation correlated with ERK1/2 activation and was reversed at higher doses of Dabrafenib as p-ERK1/2 levels declined. We analysed four additional RAFi (Vemurafenib, PLX-4720, Encorafenib and GDC-0879), comparing them with cells treated with Trametinib (a MEK1/2 inhibitor or MEKi) (Supplementary Fig. 1A, B). Trametinib inhibited p-ERK1/2 and EdU incorporation whereas all RAFi stimulated paradoxical activation of RAF and ERK1/2 signalling and four out of five RAFi also inhibited EdU incorporation (Supplementary Fig. 1A, B). ERK1/2 signalling operates within a 'sweet-spot' to control cell proliferation, with high levels of p-ERK1/2 promoting cell cycle arrest[21–23] so we initially attributed the inhibition of DNA synthesis to paradoxical RAF activation driving p-ERK1/2 beyond a level optimal for proliferation. However, close examination of the data argued against this. For example, Encorafenib was more potent than Dabrafenib at increasing p-ERK1/2 but less potent at inhibiting EdU incorporation, whilst GDC-

0879 was equipotent to Dabrafenib for increasing p-ERK1/2 but failed to inhibit EdU incorporation (Supplementary Fig. 1A, B). To test directly if ERK1/2 activation was required for RAFi-induced cell cycle arrest we combined Dabrafenib with Trametinib. Trametinib caused a concentration-dependent loss of both basal and paradoxical ERK1/2 activation (Fig. 1B). The reduction in basal EdU incorporation caused by 5 nM or 10 nM Trametinib was partially reversed by Dabrafenib-dependent paradoxical ERK1/2 activation (Fig. 1C), but Trametinib was unable to reverse the loss of EdU incorporation seen with 100nM-1μM Dabrafenib (Fig. 1C). Identical results were obtained with the chemically distinct MEKi, Selumetinib (Supplementary Fig. 1C, D). Together these results demonstrated that the RAFi-induced loss of EdU incorporation was independent of ERK1/2 activation, reflecting either ERK1/2-independent signalling by RAF or an off-target effect of RAFi.

We examined the kinetics of RAFi-induced ERK1/2 activation and inhibition of EdU incorporation (Fig. 1D–F). Dabrafenib, Vemurafenib and PLX-7284 (a chemically distinct RAFi with properties similar to Vemurafenib and PLX-4720) caused rapid ERK1/2 activation that was maximal at 1 h before declining into a sustained phase; Trametinib caused the immediate (1 hour) and sustained loss of p-ERK1/2. EdU staining revealed that RAFi-treated and MEKi-treated cells exhibited a comparable proportion of cells in S-phase for the first 8 h of treatment (Fig. 1E). However, within these 'EdU-positive' cells RAFi treatment caused a very rapid reduction in EdU staining intensity, consistent with rapid inhibition of DNA replication (Fig. 1F). This was apparent within 1 h, reached a maximum 50% inhibition by 4 h (Fig. 1F) and was also seen by flow cytometry (Supplementary Fig. 2A). This was not observed in MEKi-treated cells which exhibited a slower inhibition of DNA synthesis between 8 and 24 hours, reflecting G1 arrest. The rapid inhibition of DNA replication was reminiscent of an acute cell-cycle checkpoint such as the DNA damage response (DDR) and the kinetics of the RAFi response matched that in cells treated with Etoposide (Fig. 1F), but Dabrafenib and PLX-7284 failed to increase p-H2AX(S139) or p-CHK1(S345), both classic markers of DNA damage (Supplementary Fig. 2B–D). In summary, a variety of chemically distinct RAFi initiated a rapid, ERK1/2-independent inhibition of DNA replication in a manner reminiscent of an acute DDR but without evidence of DNA damage.

### RAF inhibitors drive rapid, ERK1/2-independent activation of the unfolded protein response (UPR)

To identify RAFi-driven, MEK1/2-ERK1/2-independent signalling pathways we performed a RNA-seq experiment. NCI-H358 cells were treated for 4 or 24 hours with 1μM Dabrafenib alone (Dab), 30 nM Selumetinib (Sel) alone or the combination (Dab+Sel); these doses were previously optimised (Supplementary Fig. 1C, D). Replicate samples with the same drug mastermix confirmed that Dab-driven activation of ERK1/2 was completely reversed by 30 nM Selumetinib (Dab+Sel) whilst Sel alone reduced basal p-ERK1/2 levels (Fig. 2A). Analysis of the RNA-seq data confirmed that Sel reversed Dab-induced expression of *DUSP4*, *DUSP5* and *DUSP6* whilst Sel alone reduced the basal expression of these three well known ERK1/2 target genes (Supplementary Fig. 3A); this showed that the experimental set-up had worked in identifying MEK1/2-dependent gene expression. Hallmark Genesets from the molecular signatures database were used to analyse the RNA-seq, looking for gene sets induced or repressed by Dab and Dab+Sel (Supplementary Fig. 3B). Dabrafenib-induced, MEK1/2-dependent gene sets such as 'KRAS signalling Up', 'Epithelial Mesenchymal Transition' and 'Inflammatory responses' also validated the experiment. Only three gene sets responded to Dab and Dab+Sel equally: 'E2F targets' and 'G2M Checkpoint' decreased strongly, likely reflecting the inhibition of DNA replication. The only gene set increased by RAFi in a MEK1/2-independent fashion was the 'Unfolded Protein Response' or UPR, a homoeostatic signalling network activated by the accumulation of damaged or misfolded proteins in the ER lumen (ER stress)[28,29] and consisting of three pathways controlled by

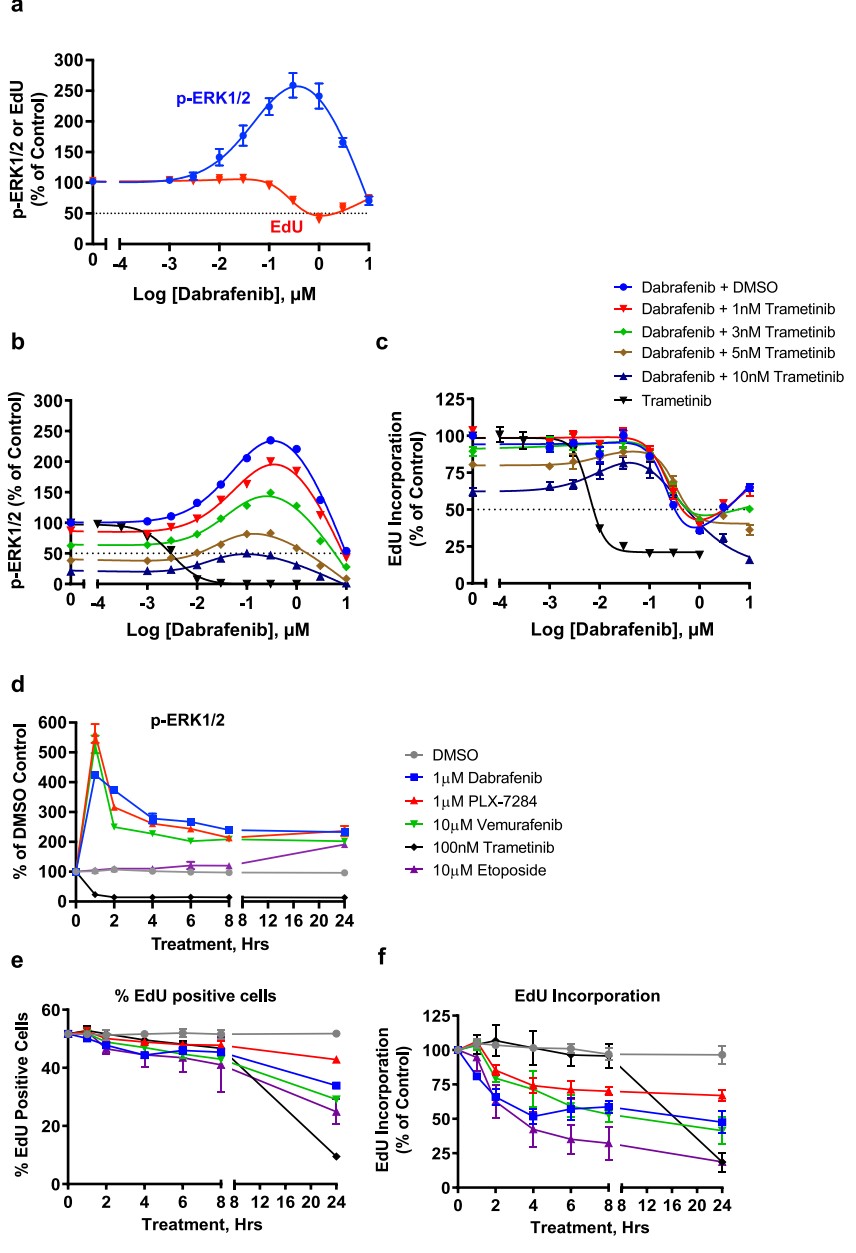

**Fig. 1 | RAF inhibitors drive the rapid inhibition of DNA replication independent of paradoxical ERK1/2 activation. a** NCI-H358 cells treated with the indicated concentrations of Dabrafenib for 24 hours with a 10 μM EdU pulse for the final hour. Cells were then fixed and permeabilized for EdU detection (red line), immunofluorescence with a p-ERK1/2-specific antibody (blue line) and co-stained with DAPI. Mean signal per cell was determined by high-content image analysis of 2000-15000 cells per condition and normalised to DMSO control. Normalised mean values ± SD are shown, $n = 3$ biological replicate experiments (**b**, **c**) NCI-H358 cells were treated with 1 (red triangle), 3 (green diamond), 5 (brown diamond) or 10 nM (dark blue triangle) Trametinib for 1 hour prior to addition of the indicated concentration of Dabrafenib for 24 hours. Some cells received increasing concentrations of Trametinib alone for 24 hours (black triangle). p-ERK1/2 (**b**) and EdU (**c**) incorporation were quantified by HCM analysis as above. Results shown are mean ± SD of $n = 2$ (pERK1/2) or 4 (EdU) biological replicate experiments. **d**–**f** NCI-H358 cells were treated with 1 μM Dabrafenib (blue square), 1 μM PLX-7284 (red triangle), 10 μM Vemurafenib (green triangle), 100 nM Trametinib (black triangle) or 10 μM Etoposide (purple triangle) for the indicated times and analysed for p-ERK1/2 (**d**), percentage of EdU-postive cells (**e**) or intensity of EdU staining (**f**) as above. Results mean ± SD of $n = 3$ biological replicate experiments.

distinct stress sensors in the ER membrane. ATF6 translocates to the Golgi apparatus where it undergoes proteolytic cleavage before entering the nucleus to drive transcription of XBP-1 and components of the ER-associated degradation (ERAD) machinery. Inositol-requiring protein 1α (IRE1α) coordinates with ATF6 by driving the splicing of RNA encoding the XBP-1 transcription factor, resulting in expression of the active XBP-1s transcription factor which drives expression of ER chaperones. Finally, Protein kinase RNA-like ER kinase (PERK) phosphorylates translation initiation factor eIF2α; this attenuates protein synthesis but allows non-canonical translation of ATF4, which in turn drives expression of CHOP and ATF3. Further analysis of the RNA-seq data confirmed strong Dab-induced, MEK1/2-independent expression of 10 canonical UPR genes including ATF4, ATF3, CEBPB, DDIT3 (CHOP), DDIT4 (REDD1) and SESN2 (Supplementary Fig. 3C). When we examined the RNA-seq replicate cell extracts (Fig. 2A) we observed a striking MEK1/2-ERK1/2-independent increase in expression of ATF4, CHOP and their downstream target TRIB3, an ER-sensing pseudokinase[30], following 4 or 24 hours Dab treatment.

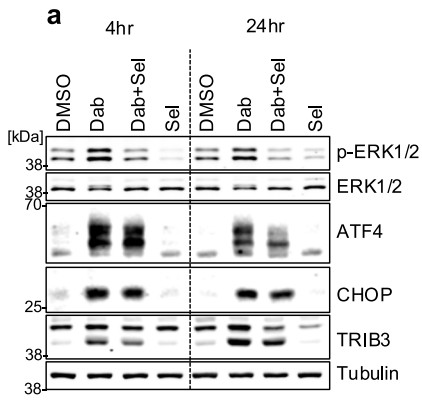

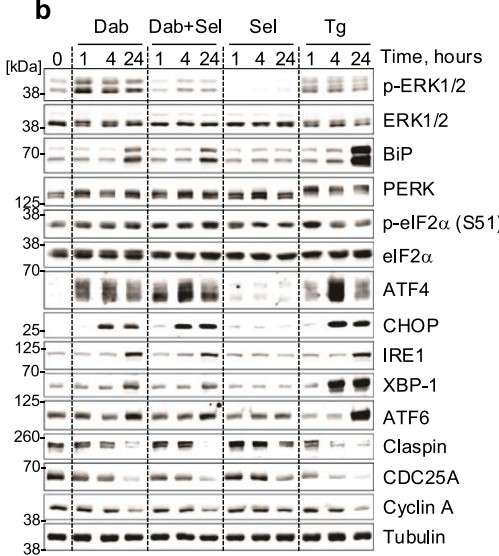

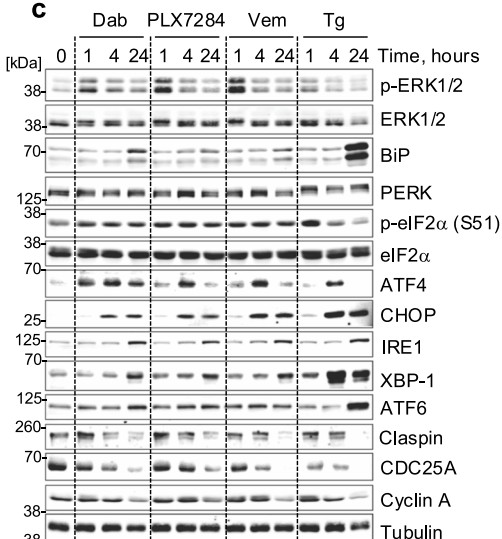

**Fig. 2 | RAFi drive rapid, ERK1/2-independent expression of ATF4 and CHOP but not other arms of the UPR. a** NCI-H358 cells were treated with 1 μM Dabrafenib (Dab), 30 nM Selumetinib (Sel) or the combination (Dab+Sel) for 4 or 24 hours. Samples were processed for RNA-seq (Supplementary Fig. 3) and replicate dishes were used to prepare whole cell extracts that were resolved by SDS-PAGE and immunoblotted with the indicated antibodies. **b** NCI-H358 cells were treated with Dab, Sel and Dab+Sel as above and additionally with 100 nM Thapsigargin (Tg) for 1, 4 or 24 hours. Whole cell lysates were fractionated by SDS-PAGE and immunoblotted with the indicated antibodies. **c** NCI-H358 cells were treated with 1 μM Dabrafenib (Dab), 1 μM PLX-7284 (PLX), 10 μM Vemurafenib (Vem), or 100 nM Thapsigargin (Tg) for 1, 4, or 24 hours. Whole cell extracts were resolved by SDS-PAGE and immunoblotted with the indicated antibodies. All results are representative of two independent experiments and in all cases results were captured by LiCor.

To investigate the UPR NCI-H358 cells were treated with Thapsigargin (Tg), an ER stress agonist, which activated PERK (indicated by its hyperphosphorylation) and increased expression of ATF4, CHOP, XBP-1 and ATF6 consistent with activation of all arms of the UPR (Fig. 2B). Dabrafenib stimulated a rapid (1 hour) increase in ATF4 and a slower increase in CHOP but little activation of PERK and only modest expression of BIP, ATF6 or XBP1. Selumetinib alone had no effect on ER stress markers but effectively inhibited basal and Dab-induced p-ERK1/2. Critically, Selumetinib did not reverse the Dab-induced expression of ATF4 or CHOP (Fig. 2B). A similar profile of ATF4 and CHOP expression but modest and delayed expression of other markers of ER stress was seen with two other RAFi, PLX-7284 and Vemurafenib (Fig. 2C). To further validate this finding, we established a HCM assay for ATF4, CHOP and XBP-1 using Tg or Tunicamycin (Tn) as positive controls (Supplementary Fig. 4A–C). This demonstrated that both RAFi, Dabrafenib and PLX-7284,

induced ATF4 expression with similar kinetics and to the same extent as Tg and Tn. However, whilst Dabrafenib and PLX-7284 increased CHOP and XBP-1 expression, the magnitude of this response was far less than that seen with Tg or Tn; CHOP and XBP-1 expression was much more responsive to classical ER stressors.

To test the involvement of PERK we used GSK2606414[31] a potent, selective PERK kinase inhibitor. This blocked the hyperphosphorylation of PERK, the expression of ATF4 and CHOP and the slower expression of ATF6 in response to Thapsigargin but had no effect on the expression of ATF4 or CHOP in response to Dabrafenib (Fig. 3A). We used HCM to quantify the effects of the PERK inhibitor; GSK2606414 inhibited CHOP expression (Fig. 3B) and reversed the inhibition of EdU incorporation (Fig. 3C) induced by Tg or Tn in a concentration-dependent fashion, indicating that PERK was required for these responses to ER stress. In contrast, the inhibition of EdU incorporation induced by Dabrafenib, Vemurafenib or PLX-7284 was

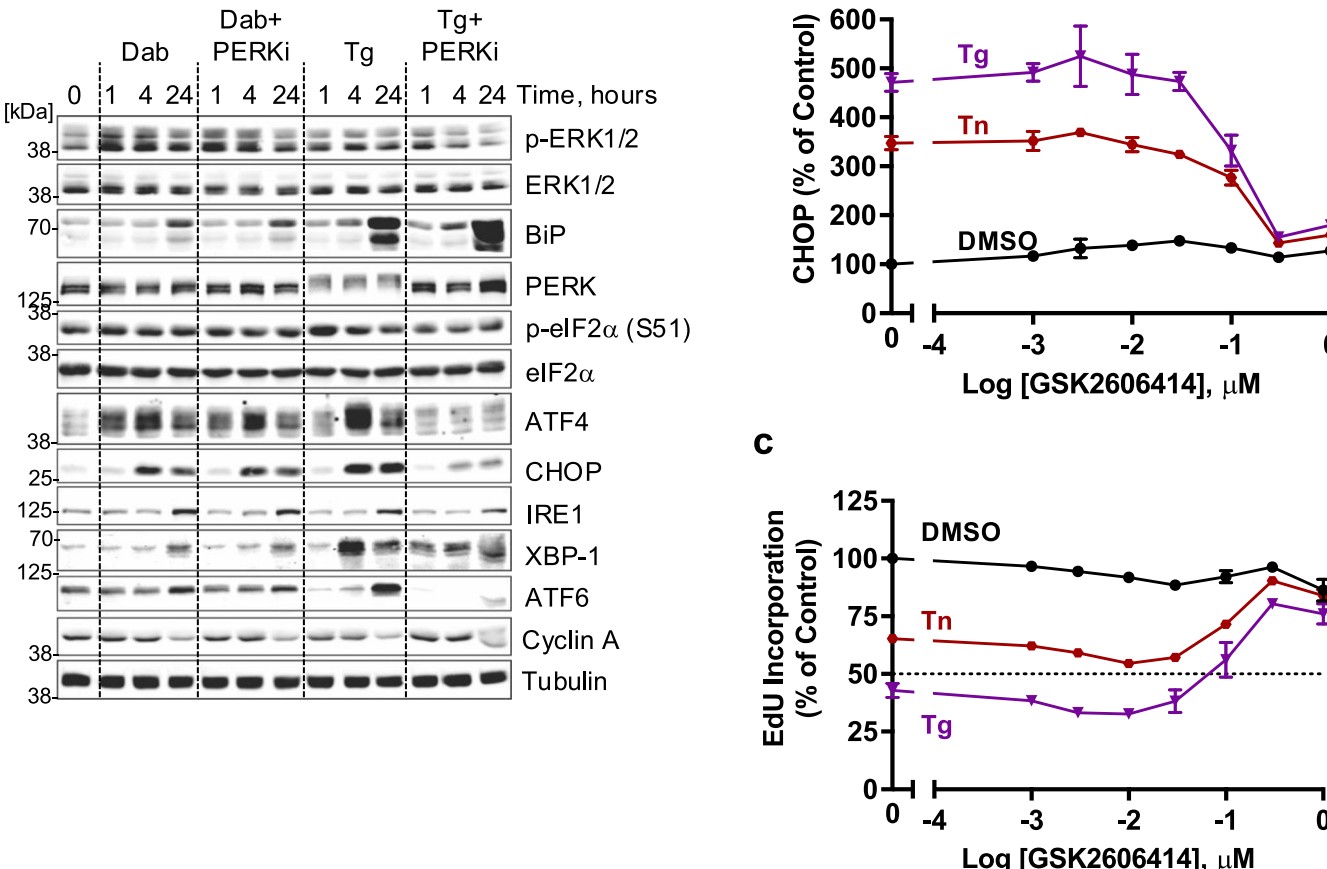

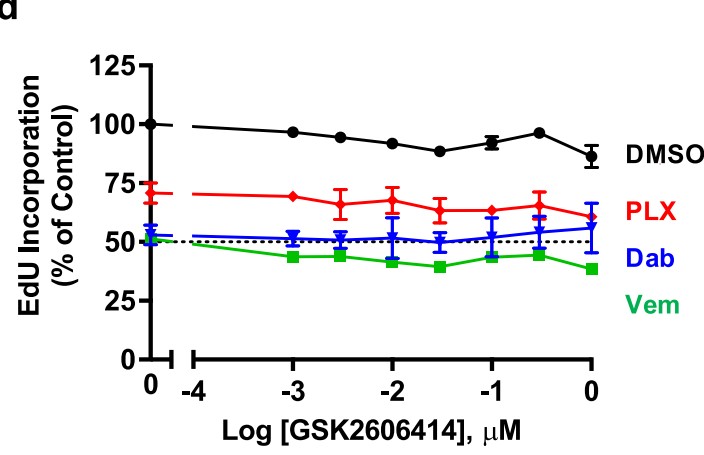

**Fig. 3 | PERK plays no role in the RAFi-induced expression of ATF4 or CHOP or inhibition of cell proliferation. a** NCI-H358 cells were pretreated for 30 minutes with either DMSO or 300 nM GSK2606414 (PERK inhibitor, PERKi) prior to the addition of 1 μM Dabrafenib (Dab) or 100 nM Thapsigargin (Tg) for 1, 4 or 24 hours. Whole cell extracts were resolved by SDS-PAGE and immunoblots with the indicated antibodies were captured by LiCor. Results are representative of three independent experiments. **b, c** High content microscopy (HCM) analysis of CHOP expression and EdU incorporation in NCI-H358 cells treated with the indicated concentrations of GSK2606414 for 30 mins prior to the addition of DMSO (black circle), 100 nM Thapsigargin (Tg, purple triangle) or 2 μg/ml Tunicamycin (Tn, red circle) for 24 hours. Cells were analysed for EdU incorporation as in Fig. 1. Mean signal per cell was quantified by high-content image analysis of 2000-15000 cells per condition and normalised to DMSO alone control. **d** NCI-H358 cells were treated with the indicated concentrations of the GSK2606414 for 30 mins before the addition of 1 μM Dabrafenib (Dab, blue triangle), 1 μM PLX-7284 (PLX, red diamond) or 10 μM Vemurafenib (Vem, green square) for 24 hours. Cells were analysed for EdU incorporation as above. For (**b–d**) normalised mean values ± SD are shown from $n = 3$ biological replicate experiments.

unaffected by GSK2606414 (Fig. 3D). In summary, these data showed that RAFi drive the rapid expression of ATF4 to the same magnitude as the ER stressors, Tg and Tn, but cause little or no activation of the other arms of the UPR. However, PERK plays no role in the RAFi-induced expression of ATF4 or CHOP or RAFi-induced inhibition of cell proliferation.

## RAF inhibitors activate the ISR and inhibit cell proliferation by activating GCN2

Phosphorylation of eIF2α at Ser51 attenuates protein synthesis but allows non-canonical translation of ATF4 to drive a homoeostatic gene expression programme, allowing cells to adapt to various forms of stress; this ancient signalling pathway is called the Integrated Stress Response (ISR)[24]. The rapid expression of ATF4 suggested that RAFi activated the ISR, but this was apparently not dependent on PERK. Three other eIF2α kinases (EIF2AKs) can phosphorylate Ser51 of eIF2α to initiate the ISR: PKR (Double-stranded RNA-dependent protein kinase) is activated by viral infection, HRI (Heme-regulated eIF2α kinase) by Haem depletion and GCN2 (General control non-depressible protein 2) by amino acid starvation. We examined a potential role for GCN2 using an antibody that detects the activating phosphorylation site at Thr899. Four different RAFi (Dabrafenib, PLX-7284, Vemurafenib and LY-3009120) increased p-T899 GCN2 and p-S51 eIF2α, increased the expression of ATF4 and CHOP (markers of the ISR) and caused a loss of CDC25A (linked to inhibition of DNA replication) (Fig. 4A and Supplementary Fig. 5A). Similar results were observed with two further RAFi, Encorafenib and PLX-8394 (Plixorafenib) (Supplementary Fig. 5A), whereas Thapsigargin failed to activate GCN2 but caused hyperphosphorylation of PERK, activation of the ISR and loss of CDC25A (Fig. 4A). The pan-RAFi, LY-3009120, inhibits all RAF isoforms so fails to drive paradoxical ERK1/2 activation but still activated GCN2, demonstrating that RAFi-dependent activation of GCN2 and the ISR was independent of ERK1/2 activation (Fig. 4A). RAFi-induced phosphorylation of GCN2 (p-T899) was inhibited by A-92[32], an ATP-competitive GCN2 kinase inhibitor (Supplementary Fig. 5A), whereas the eIF2α antagonist ISRIB[33,34] had no effect on p-T899 GCN2, consistent with eIF2α being downstream of GCN2. However, ISRIB completely reversed the RAFi-induced expression of ATF4 and CHOP and also reversed the loss of CDC25A, indicating that this was a consequence of the ISR (Fig. 4A and Supplementary Fig. 5A). We also quantified ATF4 and CHOP expression and EdU incorporation by HCM. ISRIB inhibited the expression of ATF4 and CHOP and reversed the inhibition of EdU incorporation induced by Dabrafenib, PLX-7284 or Vemurafenib (Fig. 4B) but had little if any effect on the expression of ATF4, CHOP or loss of EdU incorporation seen with Tg and Tn (Supplementary Fig. 5B) suggesting that these ER stressors exert their effects via another arm of the UPR. Finally, the GCN2 kinase inhibitor A-92 reversed the expression of ATF4 and CHOP and the inhibition of EdU incorporation induced by Dabrafenib, PLX7284 and Vemurafenib (Fig. 4C) and similar results were seen for PLX-8394, LY-3009120 and Encorafenib (Supplementary Fig. 5A). In contrast, A-92 had no effect on the response to the ER stressors Tg and Tn (Supplementary Fig. 5C). Notably, the loss of CDC25A by RAFi was reversed by A-92 or ISRIB indicating that it was a consequence of the GCN2- and ISR-dependent shutdown of protein synthesis (Fig. 4A and Supplementary Fig. 5A).

We also employed genetic knockdown or knockout of GCN2 and expanded our analysis into different cell types. siRNA-mediated knockdown of GCN2 strongly inhibited Dab-driven expression of ATF4 and CHOP in NCI-H358 cells, whilst siRNA against ATF4 strongly inhibited Dab-driven CHOP expression without affecting GCN2 abundance (Supplementary Fig. 6A-C). Dabrafenib increased expression of ATF4 in wild type immortalised MEFs (iMEFs) and PERK KO iMEFs, but this response was greatly reduced in GCN2 knockout iMEFs (Fig. 5A). Attempts to generate GCN2 KO NCI-H358 cells using CRISPR/Cas9 gene editing were unsuccessful; however, knockout of GCN2 was

successful in HCT116 colorectal cancer cells (Fig. 5B-H). In HCT116 cells Dabrafenib elicited a significant activation of GCN2 (Fig. 5B, C). GCN2 KO abolished expression of ATF4 and CHOP induced by Dabrafenib and Histidinol, an inhibitor of histidyl tRNA synthetase that mimics histidine starvation, thereby activating GCN2 through accumulation of uncharged tRNAs (Fig. 5B); in contrast, the response to Tunicamycin was unaffected by GCN2 KO. We expanded this analysis by comparing four different RAFi (Dabrafenib, Encorafenib GDC-0879 and LY-3009120) and two indirect activators of GCN2 (NXP800[35] and Histidinol). Dabrafenib, Encorafenib and LY-3009120 all increased p-T899 GCN2, p-S51 eIF2α and ATF4 and CHOP expression and these responses were almost completely GCN2-dependent (Fig. 5D-H); in contrast GDC-0879 failed to activate GCN2 or the ISR (Fig. 5D-H). Finally, we used HCM assays for p-ERK1/2 and ATF4 to screen additional RAFi using comparisons with the previous inhibitors (Supplementary Fig. 7A–D). Paradox breaker RAFis (PLX-8394 and PLX-7922) derived from Vemurafenib[20] were less potent and effective for ERK1/2 activation but still increased ATF4 expression. Both pan-RAF inhibitors, LY-3009120 and AZ628, strongly inhibited ERK1/2 activation. LY-3009120 increased ATF4 expression over the same concentration range as it inhibited p-ERK1/2; in contrast, AZ628 was very poor at increasing ATF4 despite strong p-ERK1/2 inhibition.

Taken together, these results using small molecule inhibitors (ISRIB and the GCN2 inhibitor, A-92) and genetic knockdown or knockout of GCN2 demonstrated unequivocally that 8 of 10 RAFi studied activated GCN2 and the ISR in cells and this required GCN2 kinase activity. Only GDC-0879 and AZ628 failed to activate GCN2 and/or the ISR.

**RAFi bind directly to GCN2 dimers and drive their activation in vitro and in cells.** In considering how RAFi might activate GCN2 we noted that Dabrafenib rapidly increased p-T899 GCN2 and p-S51 eIF2α (15–30 mins) and ATF4 expression (30 min-1 hour) (Fig. 6A); the increase in ATF4 expression was notably more rapid than that seen with Tunicamycin, a validated ISR agonist. Similar, rapid phosphorylation of GCN2 and expression of ATF4 was observed in Dabrafenib-stimulated colorectal cancer cells, regardless of whether they exhibited paradoxical activation of ERK1/2 (HCT116 cells, KRAS^G13D) or inhibition of ERK1/2 (HT29 cells, BRAF^V600E) (Fig. 6B). HT29 cells had much lower expression of GCN2 than HCT116 cells and exhibited lower expression of ATF4 and CHOP, but both cells responded to Dabrafenib with a rapid phosphorylation of GCN2. The rapid activation of GCN2 and ATF4 expression was reminiscent of ERK1/2 activation (apparent at 15–30 mins) arising from Dabrafenib-induced activation of wild type RAF dimers (Figs. 1D and 6A) so we tested the possibility that GCN2 activation might be RAF-dependent; for example, RAF activation by RAFi might allow RAF to phosphorylate GCN2 or interact with GCN2 to facilitate its activation. However, in western blot analysis siRNA-mediated knockdown of CRAF alone prevented Dabrafenib-induced ERK1/2 activation but not Dabrafenib-induced phosphorylation of p-T899 GCN2, p-S51 eIF2α or expression of ATF4 and CHOP (Supplementary Fig. 8A). Similar results were obtained in high content microscopy analysis, where CRAF knockdown prevented ERK1/2 activation by six different RAFi but had no effect on RAFi-induced ATF4 expression (Supplementary Fig. 8B-E). We considered the possibility that RAFi might disrupt amino acid metabolism, mimicking amino acid starvation, but the kinetics of GCN2-ISR activation seemed too rapid to support this possibility.

We performed concentration-response curves with four different RAFi in HCT116 cells. Dabrafenib, Encorafenib and LY-3002190 all activated GCN2 and ATF4 expression whereas GDC-0879 again did not (Fig. 6C). Interestingly, ATF4 expression (a strong and dynamic read-out of ISR activation) increased at 100 nM Dabrafenib and peaked at 1 μM before declining at 10 μM. Similar 'bell-shaped' concentration-response curves for ATF4 expression were observed for Encorafenib

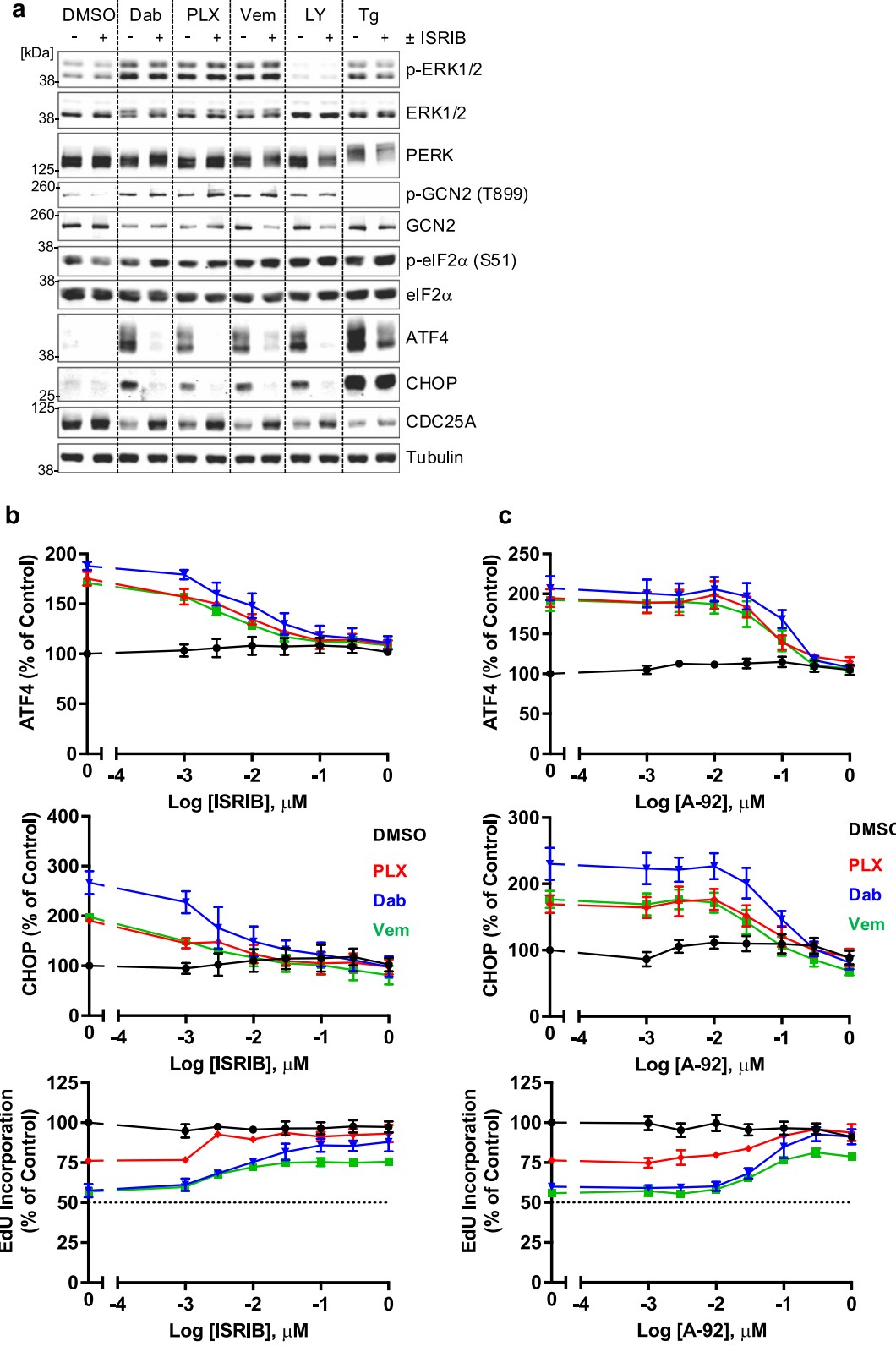

and LY-3002190 and were reminiscent of ERK1/2 activation arising from RAFi-induced activation of RAF dimers (Fig. 1A; Supplementary Fig. 1A). Previous studies showed that some early stage RAFi exhibit high affinity for the GCN2 kinase domain[36–38]. Furthermore, crystal structures of human GCN2 exhibit a back-to-back dimer configuration[38,39] consistent with the active dimer conformations of other protein kinases including PKR, IRE1 and BRAF[5,13]. Dimerization is

observed during 'normal' RAS-dependent RAF activation and is critical for paradoxical activation of RAF by RAFi[13,18]. Similarly, dimerization is critical for activation of the GCN2 kinase domain[39–41].

Prompted by our results and these precedents we considered the hypothesis that many RAFi are also GCN2 inhibitors that promote paradoxical activation of GCN2 dimers; to test this we first assessed the effect of RAFi on GCN2 in vitro. When purified, recombinant, full-

**Fig. 4 | RAF inhibitors activate ISR and inhibit cell proliferation by activating GCN2. a** NCI-H358 cells were treated with DMSO or 300 nM ISRIB for 30 mins prior to the addition of 1 μM Dabrafenib (Dab), 1 μM PLX-7284 (PLX), 10 μM Vemurafenib (Vem), 1 μM LY-3009120 (LY) or 100 nM Thapsigargin (Tg) for 4 hours. Whole cell lysates were fractionated by SDS-PAGE and immunoblotted with the indicated antibodies. Western blots from a single experiment representative of 3 independent experiments are shown. **b, c** NCI-H358 cells were treated with the indicated concentrations of either ISRIB (**b**) or A-92 (**c**) for 30 mins prior to the addition of DMSO (black circle), 1 μM Dabrafenib (Dab, blue triangle), 1 μM PLX-7284 (PLX, red diamond) or 10 μM Vemurafenib (Vem, green square). Cells were treated for 7 hours with a pulse of 10 μM EdU for the final hour, then fixed and permeabilized for EdU detection, immunofluorescence with ATF4- or CHOP-specific antibodies and co-stained with DAPI. Mean signal per cell was quantified by high-content image analysis of 2000-15000 cells per condition and normalised to DMSO control. Normalised mean values ± SD are shown, from n = 3 biological replicates (except in the case of Dabrafenib + ISRIB treatment where n = 4).

length human GCN2 dimers were incubated with increasing doses of Dabrafenib in in vitro kinase assays we observed a concentration-dependent increase in p-T899, indicating activation and autophosphorylation of GCN2 (Fig. 7A); furthermore, as with the cellular response (Fig. 6C), this response was also 'bell-shaped', peaking at 0.5-1 μM before declining at higher doses (Fig. 7A). In contrast, GDC-0879 failed to increase p-T899 GCN2 at any doses (Fig. 7B), agreeing with its failure to activate GCN2 in cells (Figs. 5D–H and 6C). Smaller concentration response curves, to allow side-by-side comparison of four RAFi, demonstrated that LY-3009120, Encorafenib and Dabrafenib activated purified GCN2 dimers at 100 nM, peaking at 1 μM and then declining at 10 μM whereas GDC-0879 again failed to activate GCN2 (Fig. 7C). In Fig. 7C we also included purified eIF2α as a substrate for GCN2 and observed that p-S51 eIF2α increased at 100 nM and 1 μM LY-3009120, Encorafenib and Dabrafenib before declining at 10 μM. In summary, these results demonstrated that LY-3009120, Encorafenib and Dabrafenib could both bind to, and activate, purified GCN2 dimers in vitro, mirroring their effects in cells and confirming a RAF-independent property for these RAFis at doses at which they activate ERK1/2 signalling.

**Structural and mutational insights into RAFi-dependent activation of GCN2.** Given the ability of RAFi to bind and activate purified, preformed GCN2 dimers, we generated structural models of GCN2-RAFi complexes. In common with BRAF, crystal structures of active human GCN2 exhibit a back-to-back dimer conformation[38,39] (Fig. 8A). Cross-dimer interactions between the side chains of Tyr651 and Tyr652 are central in the dimeric interface (Fig. 8B) and are dependent on Tyr651 occupying a 'Tyr-up' position. In contrast, these interactions are not observed in the alternative dimeric arrangement of yeast GCN2, where the side chain of Tyr651 points towards the kinase active site[42]. This 'Tyr-down' configuration blocks catalytic output in other kinases (Nek7 and IRE1) and its reversal allows back-to-back dimerization and kinase activation[43]. In an active kinase, the Tyr651 side chain forms the top R4 position in the regulatory spine (R-spine), a hallmark of active kinase structures, which comprises four hydrophobic side chains stacked in four specific positions, termed RS1-RS4, (Fig. 8B)[44,45]. Critically, the R-spine cannot assemble when Tyr651 is in a Tyr-down position because the Tyr651 side chain occupies the RS3 space, leaving RS4 empty. Dabrafenib, a Type I.5 inhibitor, is predicted to bind to GCN2 and occupy the RS3 space; this is only compatible with a Tyr651-up position, occupying RS4, stabilising an active conformation that promotes an active back-to-back dimer (Fig. 8C). At low concentrations of Dabrafenib this would inhibit one protomer but may paradoxically support activation of the second, drug-free, ATP-bound protomer potentially through dimerization-induced allostery. At higher concentrations of Dabrafenib, both kinase sites would be occupied and inhibited. In the model of GCN2 bound to LY3002190, a Type II inhibitor, the RS3 space is occupied by Leu640 and LY3002190 occupies RS2 promoting a DFG-out, C-helix in, Tyr-up conformation (Fig. 8D), assembling the R-spine to promote an active back-to-back dimer which supports paradoxical activation. In contrast, GDC-0879, a Type I inhibitor, does not occupy the RS3 space so the R-spine does not assemble. GCN2 has a bulkier gatekeeper residue

(Met) compared with BRAF (Thr), and GDC-0879 binding is predicted to dock in an orientation that fails to stabilise an active conformation (Fig. 8E).

These structural predictions are entirely consistent with the results we observed for RAFi activation of GCN2 and the ISR in cells but were predicated on the assumption that the RAFi bind to the GCN2 canonical kinase domain to elicit its activation. However, full length GCN2 contains both a conventional kinase domain and a pseudokinase domain[46] which could feasibly support RAFi binding and drive allosteric (kinase-independent) signalling through conformational changes in signalling complexes. The ability of A-92, an ATP-competitive GCN2 inhibitor, to inhibit RAFi-induced ISR activation (Fig. 4 and Supplementary Fig. 5A) suggested a requirement for the canonical kinase domain, so we generated a catalytically inactive (kinase dead, or KD) mutant EGFP-GCN2[K619A] and transiently expressed this in GCN2 KO HCT116 cells, comparing outputs with wild type EGFP-GCN2. WT GCN2 restored Dabrafenib-induced autophosphorylation of GCN2 at T899, whereas the GCN2 KD mutant failed to do so (Fig. 9A). Protein kinases possess a 'gatekeeper' residue whose molecular identity can dictate the binding affinity of ATP-competitive kinase inhibitors[47]. Mutation of this site can also influence inhibitor binding, basal kinase activity and confer drug sensitisation or resistance to various compounds[12,48,49]. In BRAF the gatekeeper residue is Thr529 whereas in GCN2 it is the bulky Met802. We generated and transiently expressed two very different GCN2 gatekeeper mutants, EGFP-GCN2[M802A] or EGFP-GCN2[M802F], in GCN2 KO HCT116 cells, using WT EGFP-GCN2 as a control. We used NXP800[35], an indirect activator of GCN2, to verify that our gatekeeper mutants were still functional. Treatment of cells expressing WT or EGFP-GCN2[M802A] (Fig. 9A) with NXP800 caused an increase in p-T899 indicating activation and autophosphorylation of GCN2. Unexpectedly, treatment of cells expressing EGFP-GCN2[M802A] with 1 μM Dabrafenib caused only a modest decrease in p-T899 indicating that the kinase was directly inhibited at this concentration. This suggested that mutation of this site may have altered the binding affinity of the drug. To test this hypothesis, we transiently expressed WT or M802A GCN2 and treated the cells with a range of concentrations of Dabrafenib. Remarkably, EGFP-GCN2[M802A] was maximally activated at 1–3 nM Dabrafenib as judged by p-T899 and increased expression of CHOP, compared to WT which was maximally activated at 0.3-1 μM (Fig. 9B). Thus introduction of a smaller gatekeeper residue facilitated Dabrafenib binding, resulting in GCN2 activation at markedly lower doses; this directly confirmed that Dabrafenib was binding to the canonical kinase domain to activate GCN2, rather than the pseudokinase domain. Interestingly, the even bulkier gatekeeper mutant, EGFP-GCN2[M802F], acted as a constitutively active kinase, driving strong p-T899 even in the absence of Dabrafenib (Fig. 9A). Structural modelling of the GCN2[M802F] mutant (Fig. 8F) suggested that the bulkier Phe802 causes a substantial clash with the Tyr-down position of Tyr651, indirectly stabilising the active Tyr-up position through direct interactions with Leu640 in the RS3 spine space and thereby promoting formation of an active back-to-back dimer.

As we have demonstrated that binding of RAFi to the ATP-binding site of GCN2 can lead to activation of the dimer we postulated that binding of GCN2 inhibitors might also show this effect. We analysed

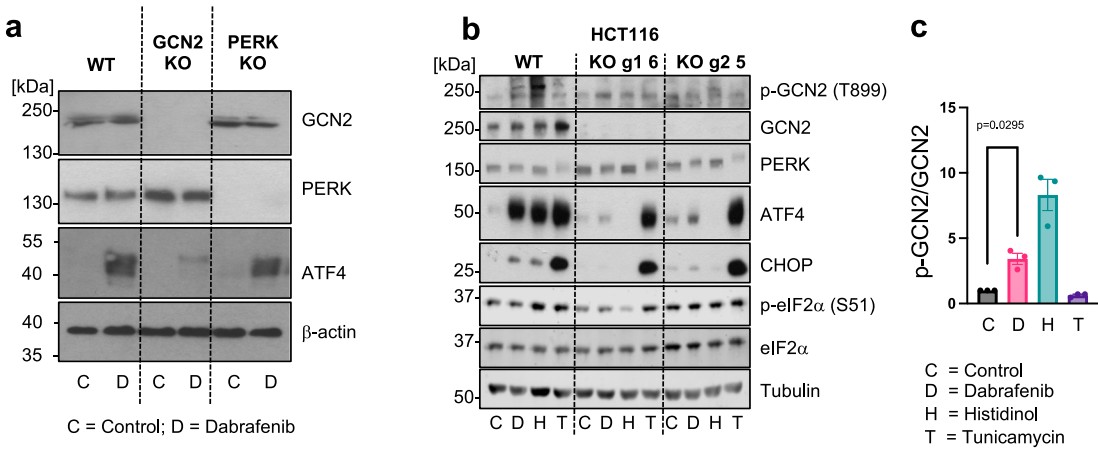

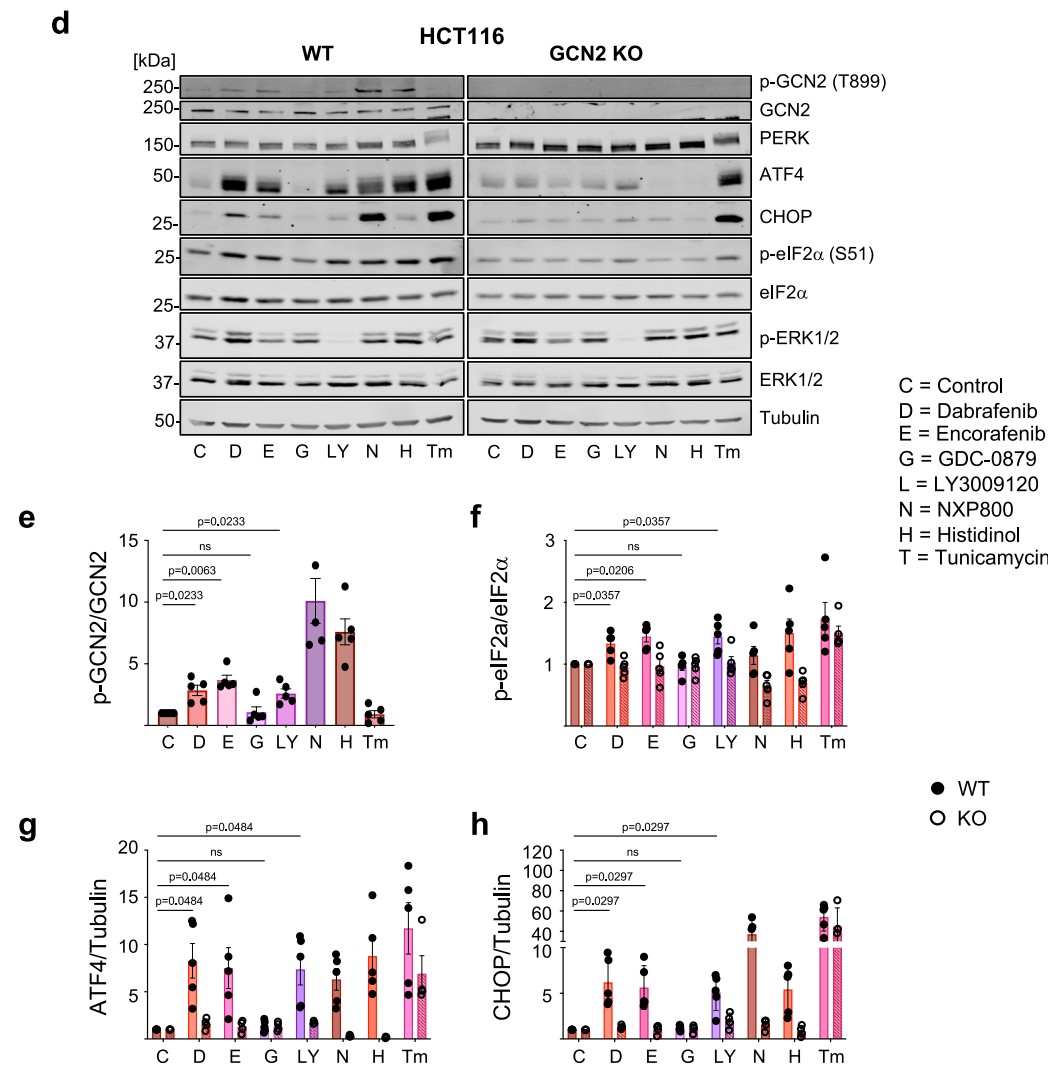

the effects of two GCN2 inhibitors, GCN2iB[50] and A-92, which we had previously used as a GCN2 inhibitor (Fig. 4C, Supplementary Fig. 5A). GCN2iB alone caused a striking concentration-dependent increase in ATF4 expression that peaked at 30 nM and started to decline at 100 nM. This response was completely lost in GCN2 KO cells indicating that GCN2iB was driving paradoxical activation of GCN2 (Fig. 10A). In contrast, A-92 failed to increase p-T899-GCN2 or ATF4 expression. To

understand the contrasting effects of these two GCN2 inhibitors, we again generated structural models. GCN2iB was predicted to bind to GCN2 (Fig. 10B) in a remarkably similar manner to Dabrafenib (Fig. 8C); both occupy the RS3 space, stabilise Tyr651 in a Tyr-up position and assemble the R-spine to promote an active back-to-back dimer. In contrast, A-92 is a Type I inhibitor (like GDC-0879); in two different binding 'poses' it failed to reach into the RS3 space so the R-spine is not

**Fig. 5 | Knockout of GCN2 blocks RAFi-induced activation of the ISR.**
**a** Immortalised wild type, GCN2 KO or PERK KO mouse embryonic fibroblasts were treated with DMSO (C, control) or 1 μM Dabrafenib (D) for 4 hours. Whole cell lysates were fractionated by SDS-PAGE and immunoblotted with the indicated antibodies, a representative experiment is shown (*n* = 3 biological replicates)
**b** Wildtype HCT116 or 2 independent CRISPR generated GCN2 knockout (KO) clones; guide 1 clone 6 (g1 6) or guide 2 clone 5 (g2 5) were treated with 1 μM Dabrafenib (D), 16 mM Histidinol (H) or 2 μg/ml Tunicamycin (T) for 4 hours. Immunoblots were probed with the indicated antibodies. **c** The ratio of p-GCN2 to total GCN2 was quantified and normalised to DMSO control (C, grey bar). Relative values ± SEM are shown, from *n* = 3 biological replicate experiments, *p* values were determined by one sample (two tailed) *t* test (refer to source data file). (Dabrafenib (D) = pink bar, Histidinol (H) = turquoise bar, Tunicamycin (T) purple bar) (**d**). Wild

type or GCN2 KO (clone g1 6) HCT116 cells were treated with either DMSO (C, control), 1 μM Dabrafenib (D), 3 μM Encorafenib (E), 1 μM GDC-8079 (G), 1 μM LY-3009120 (LY), 0.3 μM NXP800 (N), 16 mM Histidinol (H) or 2 μg/ml Tunicamycin (T) for 4 hours and immunoblotted with the indicated antibodies. Results were captured by LiCor and a representative experiment is shown. WT and KO lysates were run on separate gels but gels/blots were processed in parallel and loading controls are shown. **e-h** Ratio of p-GCN2 to total GCN2, peIF2α to total eIF2α, ATF4 to tubulin and CHOP to tubulin were quantified from blots and normalised to the relevant DMSO control. Different treatments are shown in different coloured bars, shaded bars represent KO. Relative values ± SEM are shown, from *n* = 5 (WT = black circle) *n* = 4 (GCN2 KO = empty circle) biological replicate experiments and statistical analysis was performed as in (**c**) and *p* values were adjusted according to the Holm-Šídák method (refer to source data file).

---

assembled and GCN2 is not activated (Fig. 10C, D). Thus, GCN2iB, like Dabrafenib, can act as a GCN2 activator at low doses before inhibiting GCN2 dimers at high doses, whereas at doses up to 1 μM A-92 acts predominantly as a GCN2 inhibitor. This additional insight explains why A-92 was effective as a GCN2 inhibitor and prevented Dabrafenib-induced ISR activation (Fig. 4C and Supplementary Fig. 5A).

## Discussion

Whilst studying paradoxical activation of RAF we found that the majority of RAF inhibitors simultaneously drive activation of GCN2 in cells, resulting in phosphorylation of eIF2α and activation of the ISR (Supplementary Fig. 9). We discovered this due to the unexpected ability of RAFi to inhibit DNA replication accompanied by loss of Claspin and CDC25A (Fig. 2B, C) two key regulators of the DNA damage response[51]. However, RAFi did not activate markers of DNA damage such as pS139-H2AX or p-S345 CHK1, the kinase that normally promotes degradation of CDC25 following DNA damage. MEK1/2 inhibition prevented ERK1/2 activation but did not reverse the antiproliferative effect of RAFi. Rather, RNA-seq revealed that RAFi treatment led to the ERK1/2-independent activation of the Unfolded Protein Response. Ultimately, comparisons with ER stress agonists and use of PERK inhibitors or PERK KO cells ruled out the UPR per se and allowed us to focus on the ISR. Knockdown, KO or inhibition (by A-92) of GCN2 and antagonism of p-eIF2α (by ISRIB) prevented the RAFi-induced expression of ATF4 and CHOP; notably, A-92 and ISRIB also reversed the loss of CDC25A and the inhibition of DNA replication indicating that the antiproliferative effects of RAFi likely reflect activation of the ISR and attenuation of translation.

To investigate the underlying mechanisms we focused on Dabrafenib, Encorafenib, LY-3009120 and GDC-0879. In cells GCN2 activation was apparent at 100 nM Dabrafenib, Encorafenib or LY-3009120 and peaked at 1 μM. ATF4 expression, a highly dynamic biomarker for ISR activation, increased over the same concentration range before reducing at 10 μM, exhibiting a bell-shaped or Gaussian concentration-response curve. The same response was seen for activation of full-length, purified dimers of recombinant human GCN2 in vitro indicating that these inhibitors were binding directly to GCN2 to activate it. In contrast, GDC-0879 failed to activate GCN2 in cells or in vitro and served as a useful control. Mutation of the gatekeeper residue to a smaller alanine (GCN2^M802A) in the ATP-binding pocket sensitized GCN2 to Dabrafenib in cells, confirming that RAFi bind to the canonical kinase domain of GCN2 and not the pseudokinase or HisRS domains, which harbour cryptic ATP-binding sites. Whilst expression of WT GCN2 or GCN2^M802A reconstituted RAFi-induced ATF4 and CHOP expression in GCN2 KO cells, the kinase-dead mutant (GCN2^K619A) failed to do so, indicating that binding to the ATP-binding pocket, and catalytic activation of the GCN2 kinase domain is required for RAFi to activate the ISR.

These results are highly reminiscent of the well-known paradoxical activation of wild type RAF dimers by RAFi[52]. In common with

WT RAF proteins GCN2 functions as a back-to-back dimer[39,40]. Our modelling suggests that GCN2 cross-dimer interactions between the side chains of Tyr651 and Tyr652 are central in the dimeric interface with RAFi-induced kinase activation dependent on Tyr651 adopting the 'Tyr-up' position to occupy RS4, assembling the R-spine and promoting back-to-back dimerization and activation[43]; such allosteric activation via the αC/β4 region is very common in kinases[53,54]. Our structural models of GCN2-RAFi complexes revealed that both Dabrafenib and LY-3009120 bind to GCN2 in a manner that assembles the R-spine with Tyr651 in the up position; in contrast, GDC-0879 did not, consistent with its failure to activate GCN2 and the ISR in cells. We also observed that low doses of GCN2iB drove paradoxical activation of GCN2 and the ISR whereas another GCN2 inhibitor, A-92, did not. Our structural models suggest that GCN2iB binds to GCN2 in a very similar manner to Dabrafenib to assemble the R-spine whereas A-92 does not. This likely explains why A-92 acted as a GCN2 inhibitor to prevent RAFi-induced activation of the ISR in our studies (Fig. 4C). These results confirm and extend a previous report that low doses of GCN2iB could activate GCN2 and this could be inhibited by A-92[55]. Based on these results A-92 might seem to be the preferred choice for inhibition of GCN2 in cell-based studies, but this is made more complex because A-92 has also been shown to activate PERK in cells[56]. However, A-92-driven PERK activation was only observed at very high doses of A-92 (10-40 μM), far in excess of the doses used herein to inhibit GCN2. Indeed, the high degree of homology between the EIFA2Ks is such that PERK inhibitors can also activate GCN2 in cells and in vitro, as can the PKR inhibitor C16[56].

Based on our results and the precedent of RAFi driving paradoxical activation of RAF dimers we propose that at low concentrations RAFi (or GCN2iB) bind to one protomer of GCN2 and inhibit it, but in doing so they drive allosteric activation of the drug-free, ATP-bound dimer partner through the interplay between Tyr651 and Tyr652 at the dimeric interface. We suggest the allosterically activated, drug-free, ATP-bound dimer partner is responsible for phosphorylating the drug-bound, inhibited protomer at T899 'in trans'. However, we cannot rule out an alternative model in which Raf inhibitor-bound GCN2 is more efficiently phosphorylated by uninhibited GCN2; indeed, these two scenarios are not mutually exclusive. Regardless of these variations on the model, subsequent dissociation of RAFi from phosphorylated GCN2 (by competition with ATP) would then allow it to phosphorylate eIF2α. This is consistent with in vitro GCN2 kinase data (Fig. 7C) where robust RAFi-driven phosphorylation of eIF2α was only observed at 100 μM ATP, consistent with ATP competing RAFi off p-T899-GCN2, allowing it to phosphorylate its substrate. Attempts to formally test this model using mutations that disrupt GCN2 dimerization are hampered by the fact that such mutations destabilise expression of full length GCN2. Nonetheless, the bell-shaped concentration-response for GCN2 activation in vitro and ATF4 expression in cells, the mutagenesis of the GCN2 kinase domain and expression in GCN2 KO cells and our structural modelling are all entirely consistent

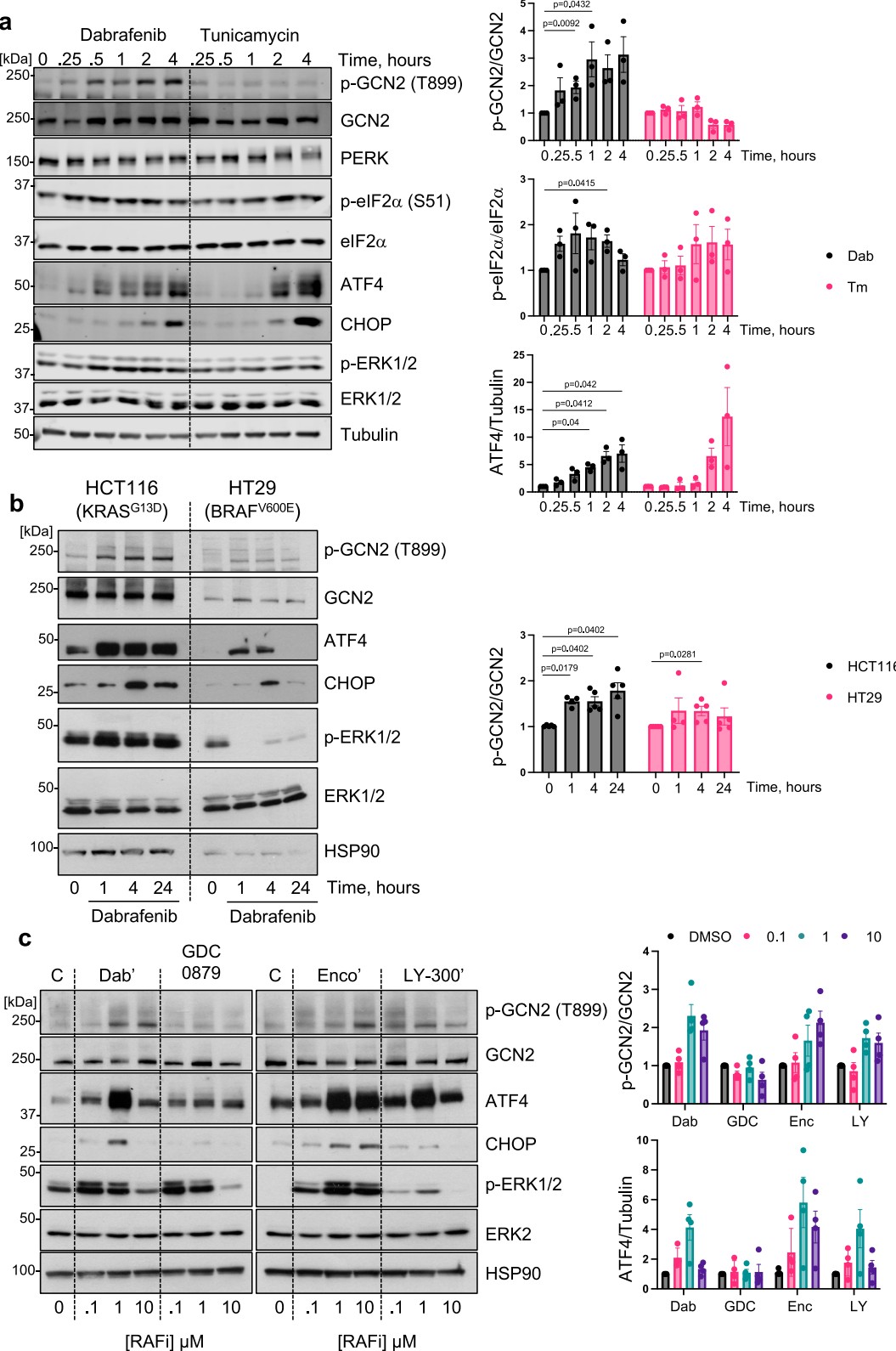

with this hypothesis. In addition, our collaborators have employed HDX-MS to analyse GCN2 activation by RAFi and GCN2iB[57], showing that Dabrafenib, Encorafenib and GCN2iB all bind within the kinase domain of GCN2, causing similar large reductions in solvent exchange in peptides surrounding the ATP-binding pocket, the activation loop, the DFG loop and T904 auto-phosphorylation site. Notably, whilst both GCN2 and RAF function as dimers, GCN2 is much larger and more complex than the RAF proteins and contains several other domains involved in inhibitory interactions that must be disengaged to drive activation[41]. So whether activation of GCN2 through the back-to-back interface is sufficient for full activation, as is seen with RAF dimers and RAFi, remains to be seen. HDX-MS analysis GCN2 activation also revealed that RAFi or GCN2iB binding propagate allosteric changes beyond the kinase domain[57]; this perhaps suggests that RAFi or

**Fig. 6 | RAF inhibitors activate GCN2 and the ISR rapidly with a bell-shaped concentration response curve. a** HCT116 cells were treated with 1 μM Dabrafenib (Dab) or 2 μg/ml Tunicamycin (Tm) for up to 4 hours. Whole cell extracts were resolved by SDS-PAGE and immunoblotted with the indicated antibodies. Results were captured by LiCor and the ratio of p-GCN2 to total GCN2, p-eIF2α to total eIF2α and ATF4 to tubulin were quantified and normalised to DMSO controls (Dab = grey bars, Tm = pink bars). Relative values ± SEM are shown, from n = 3 biological replicate experiments. *p* values were determined by one sample (two tailed) t test and were adjusted using the Holm-Šídák method (refer to source data file). **b** HCT116 or HT29 cells were treated with 1 μM Dabrafenib for up to 24 hours and whole cell lysates were analysed as above and the ratio of p-GCN2 to total GCN2 was quantified and is represented in the corresponding bar graph (HCT116 grey

bars and HT29 = pink bars), relative values ± SEM are shown (n = 5 biological replicate experiments, except 1 hr time point where n = 4), statistics were performed as in (**a**). (refer to source data file). **c** HCT116 cells were treated with either DMSO (C, control) or the indicated concentrations of Dabrafenib (Dab'), GDC-8079 (GDC), Encorafenib (Enco') or LY-3009120 (LY-300') and western blots performed as in (**a**). Experiment representative of n = 4 biological replicate experiments. Lysates were run on 2 separate gels but gels/blots were processed together. Loading controls (HSP90) are shown and quantification was relative to a DMSO control (C) run on both gels. The ratio of p-GCN2 to total GCN2 and ATF to loading control was quantified and is represented in the corresponding bar graph, relative values ± SEM are show. (Grey bars = DMSO control, pink bars = 0.1 μM, turquoise bars = 1 μM and purple bars = 10 μM of the indicated RAFi).

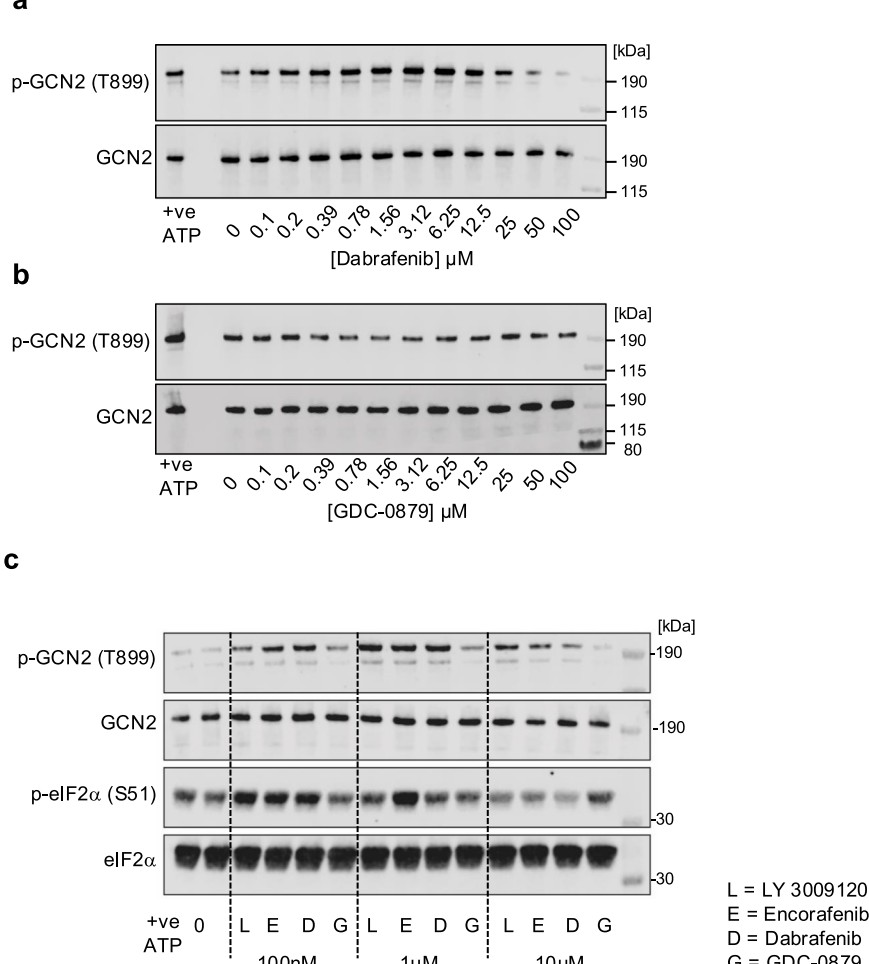

**Fig. 7 | RAF inhibitors bind directly and activate GCN2 in vitro with a bell-shaped concentration response curve. a, b** 10 nM purified recombinant full length human GCN2 was incubated at 30 °C for 20 mins with 10 μM ATP and the indicated concentrations of Dabrafenib (**a**) or GDC-0879 (**b**). Samples were resolved by SDS-PAGE and probed with the indicated antibodies. **c** Reactions were

performed as above with using 100 μM ATP and 100 nM, 1 μM or 10 μM of either LY-3009120 (L), Encorafenib (E), Dabrafenib (D) or GDC-0879 (G) and also included 2 μM eIF2α. Samples were resolved by SDS-PAGE and immunoblotted with the indicated antibodies. Single experiments representative of n = 3 biological replicates are shown.

GCN2iB binding in the kinase domain may promote a pseudo-active conformation that initiates further allosteric changes to drive full activation.

Prior studies reported that Dabrafenib[58] and Vemurafenib[59] could drive ATF4 expression but the underlying mechanism was not defined. In this study, we report activation of GCN2-ISR with 8 out of 10 chemically-distinct RAFi, suggesting that this is a property common to RAFi as a group, rather than an isolated 'off-target' effect of a single RAFi. Both the clinically approved Group 1 RAFi (Vemurafenib,

Dabrafenib and Encorafenib) and Group 2 and paradox breaker RAFi activated the ISR to the same extent as ER stress (PERK activators), histidine starvation or NXP800 (indirect GCN2 activators); only GDC-0879 and AZ628 failed to do so. Furthermore, we observed RAFi-driven activation of GCN2-ISR in tumour cells with wild type BRAF, mutant BRAF, mutant KRAS and immortalised MEFs with no ERK pathway mutation. Moreover, activation of GCN2 is not confined to EIF2AK inhibitors or RAFi; in the course of our work certain EGFR tyrosine kinase inhibitors including Erlotinib and Neratinib have also

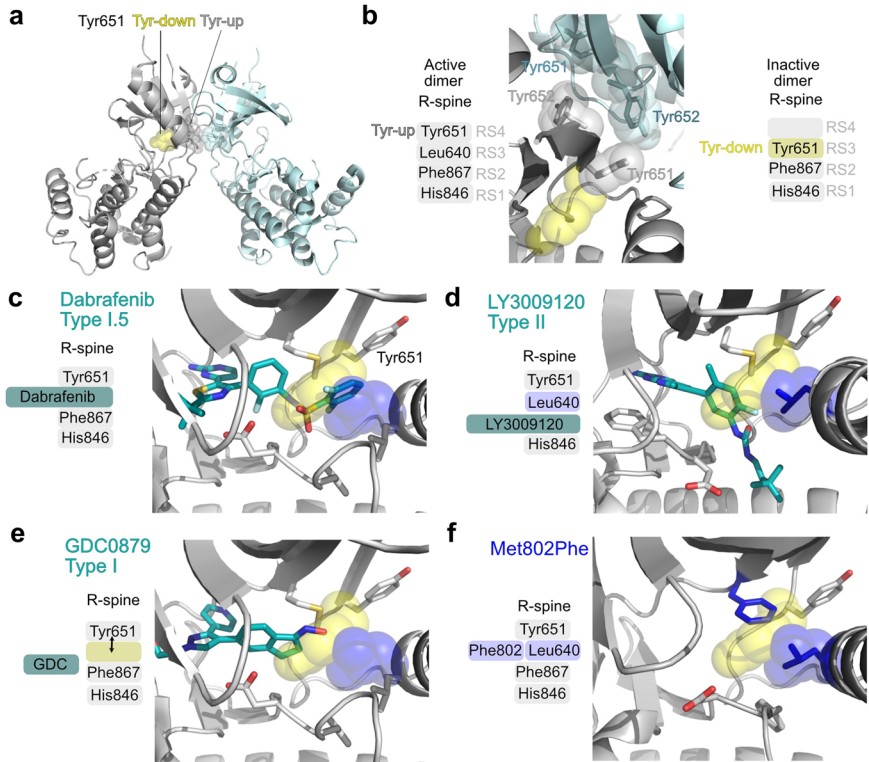

**Fig. 8 | Structural models of RAF inhibitors binding to GCN2.** Structural models of GCN2-ligand complexes were generated to provide a rationale for their biochemical properties. **a** Crystal structures of human GCN2 exhibit a back-to-back dimer, consistent with the active dimer conformations of PKR, IRE1 and other kinases. **b** Interactions between the side chains of Tyr651 (grey) and Tyr652 (blue) are central in the dimeric interface. The Tyr side chain forms the top position in the regulatory spine (R-spine), a hallmark of active kinase structures comprising four hydrophobic side chains that occupy specific positions (RS1-RS4). The R-spine cannot assemble when Tyr651 is in a Tyr-down position because the Tyr side chain occupies the space of RS3 and RS4 is empty. The Tyr-down position and RS3 space are shown as yellow and blue space filling spheres, respectively. **c** Dabrafenib binding to GCN2 is predicted to occupy the RS3 space and Tyr-651 is in the up position (**d**). The structural modelling of LY3002190 is consistent with a DFG-out, C-helix in, Tyr-up conformation with Leu640 occupying the RS3 space. **e** GDC0879, a Type I inhibitor, does not protrude into the RS3 space, which is left unoccupied; this orientation is not predicted to stabilise an active conformation. **f** Mutation of Met802 to a bulkier Phe residue is predicted to have a substantial clash with the Tyr-down position, stabilising Tyr-up through direct interactions with Leu640 in the RS3 space. See text for details.

been shown to activate GCN2 in glioblastoma cell lines[60]. However, in this case the doses of EGFRi required for GCN2 activation were substantially in excess of that required to inhibit EGFR. This contrasts with the effects reported here, where the majority of RAFi tested, including Dabrafenib, Encorafenib and Vemurafenib, activated GCN2 and the ISR over the same concentration range at which they activated RAF and the ERK1/2 pathway. This raises important questions about the extent to which ISR activation influences the clinical response to BRAF inhibitors. In cancers harbouring BRAF[V600E/K] ISR activation will be coincident with ERK1/2 inhibition, itself a pro-survival pathway[61], and ISR activation may be a route to RAFi resistance. In cells lacking BRAF mutations coincident ERK1/2 and GCN2-ISR activation may contribute to adventitious benign tumour growth. More generally, it suggests that non-ATP site pocket inhibitors of RAF such as dimerization modulators might be more desirable to avoid paradoxical activation of GCN2 and the ISR as well as RAF and the ERK1/2 pathway.

## Methods
### Reagents and Cell Lines
The source of all reagents and cell lines utilised are detailed in Table S1. Cells were grown in DMEM (HCT116) RPMI1640 (NCI-H358 and HT29) media supplemented with 10% (v/v) foetal bovine serum, penicillin (100 U/mL), streptomycin (100 mg/mL) and 2 mM glutamine. Cells were incubated in a humidified incubator at 37 °C and 5% (v/v) CO2. All cell lines were authenticated by Short Tandem Repeat (STR) profiling and confirmed negative for mycoplasma prior to experiments commencing. All reagents were from Gibco, Thermo Fisher Scientific (Paisley, UK).

### High content microscopy[26,27]
Cells were cultured in CellCarrier-96 plates (PerkinElmer). After the drug treatment, cells were harvested, fixed with 4% (v/v) formaldehyde/PBS and permeabilised with 0.2% Triton X-100 for 10 minutes. Cells were blocked for 1 hour with 2.5% goat serum (v/v) in 2% BSA/PBS (v/v) at room temperature. Cells were then incubated with primary antibody diluted in 2% BSA/PBS overnight at 4 °C. Background control wells were treated with 2% BSA/PBS, without the addition of primary antibody. Cells were washed three times with PBS and then incubated with Alexa Fluor secondary antibodies (1:500) and DAPI (1 ug/ml) in 2% BSA/PBS for 1 hour. Cells were washed with PBS before imaging using an INCELL Analyzer 6000 Microscope (GE Healthcare) imaging six fields per well. Image analysis to determine the mean signal intensity was performed using INCELL Analyzer software (https://www.gelifesciences.com/en/us/shop/cell-imaging-and-analysis/high-content-analysis-systems/software/in-cell-investigator-software-p-00344). For Edu analysis, cells were treated as described in the figure legends and 1 hour prior to harvest incubated with 10 µM 5-ethynyl-2-deoxyuridine (EdU; Click-iT EdU HCS Kit, ThermoFisher, Loughborough, UK). Fixed and processed as above and EdU detected according to the manufacturer's protocol.

### EdU incorporation by flow cytometry
Cells were treated as described in the figure legends and 1 hour prior to harvest incubated with 10 µM 5-ethynyl-2-deoxyuridine (EdU; Click-iT EdU Flow Cytometry Kit, ThermoFisher, Loughborough, UK). Cells were harvested by trypsinisation and fixed with 4% paraformaldehyde/PBS for 10 min at room temperature. EdU was detected following the

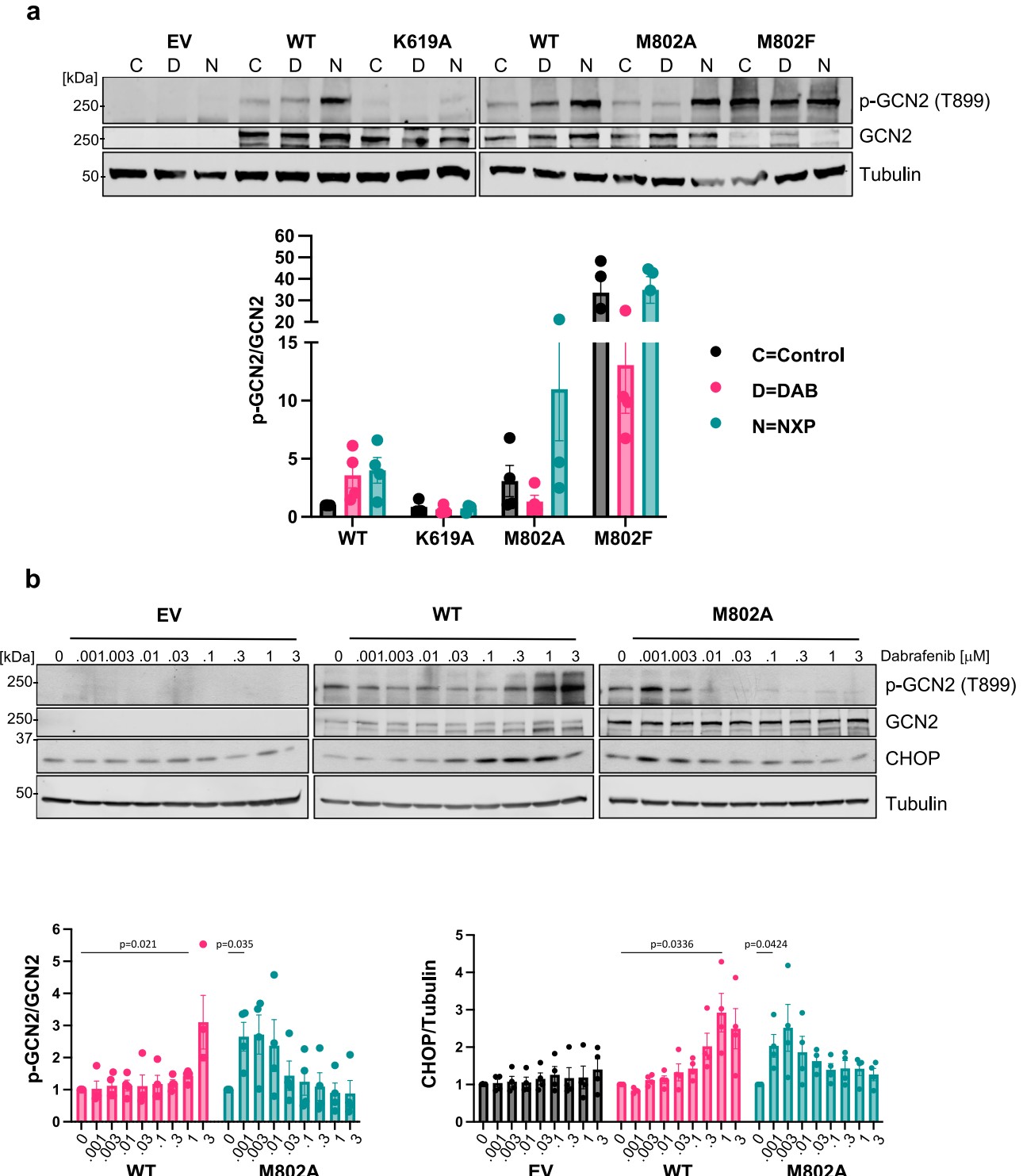

**Fig. 9 | Kinase dead and Gatekeeper mutations abolish Dabrafenib-induced activation of GCN2 and the ISR. a** HCT116 GCN2 knockout (KO) cells (clone g1 6) were transiently transfected with either empty vector (EV, EGFP-C3) or constructs expressing wildtype (WT) EGFP-GCN2, a kinase dead mutant (K619A) or one of two different gatekeeper mutations (M802A or M802F). After 48 hours, cells were treated with either DMSO (C, grey bars), 1 μM Dabrafenib (D, pink bars) or 0.1 μM NXP800 (N, turquoise bars) for 4 hours. Whole cell lysates were separated by SDS-PAGE and immunoblotted with the indicated antibodies and the ratio of p-GCN2 to total GCN2 was quantified from blots and normalised to WT DMSO control. Relative values ± SEM are shown from *n* = 4 biological replicate experiments. **b** HCT116 GCN2 KO cells were transfected with either empty vector (EV, grey bars), wildtype (WT, pink bars) or the M802A gatekeeper mutant (turquoise bars); cells were treated with the indicated dose of Dabrafenib for 4 hours. Western blots and quantification were performed as above, except that each individual construct was normalised to its own DMSO control (0). Relative values ± SEM are shown, from *n* = 4 biological replicate experiments, *p* values were determined by one sample (two tailed) *t* test (see source data file).

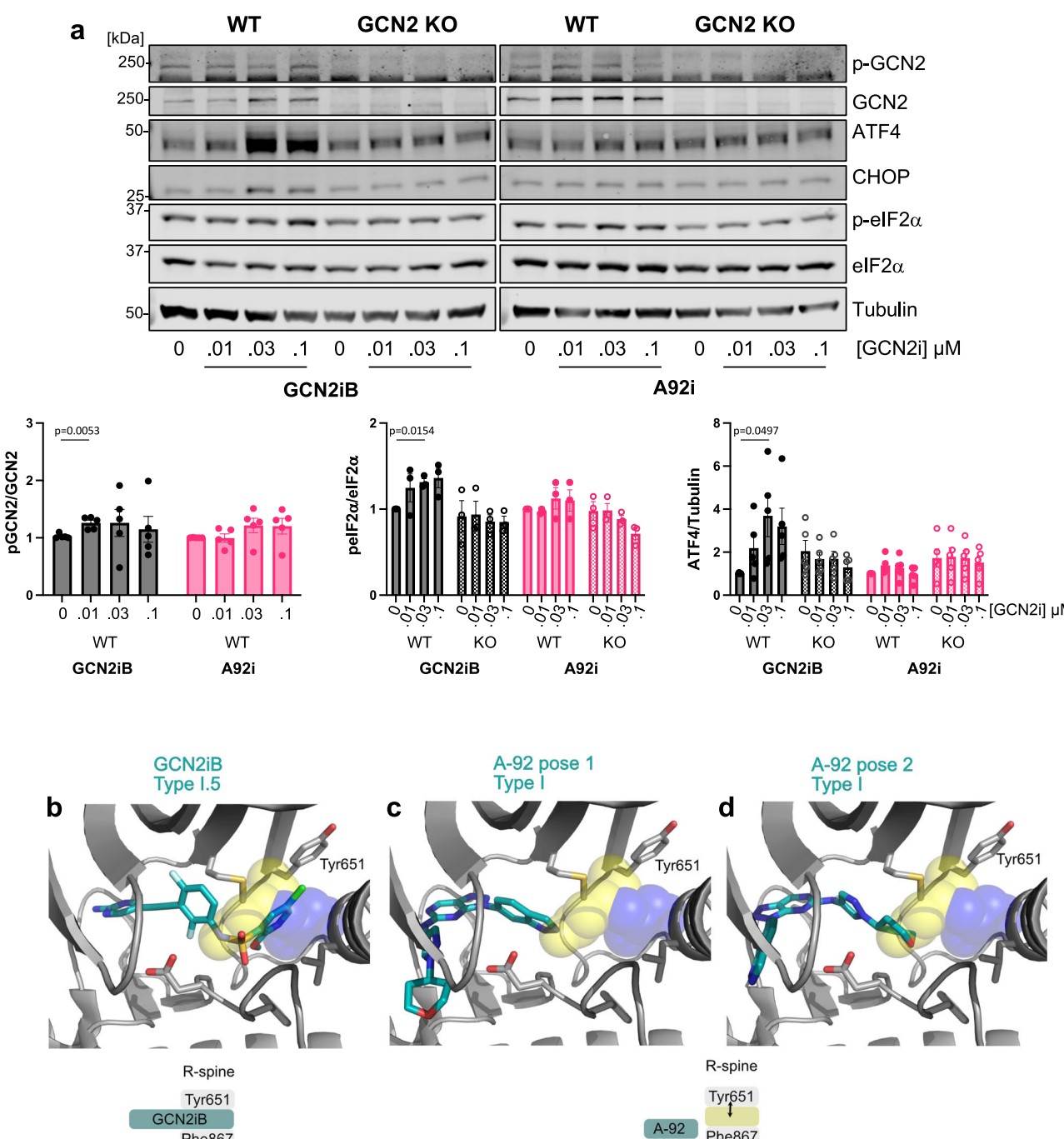

**Fig. 10 | The GCN2 inhibitor GCN2iB drives paradoxical activation of GCN2 whereas A-92 does not. a** Wildtype (WT) or GCN2 knockout (KO) HCT116 cells were treated with the indicated concentrations of either GCN2iB or A-92 for 4 hours. Whole cell lysates were separated by SDS-PAGE and immunoblots were probed with the indicated antibodies. Blots from a single representative experiment are shown. Ratio of p-GCN2 to total GCN2 ($n = 5$), peIF2α to total eIF2α ($n = 3$) and ATF4 to tubulin ($n = 5$) were quantified from blots and normalised to WT DMSO controls. Relative values ± SEM are shown from biological replicate experiments. *p* values were determined by one sample (two tailed) *t* test (refer to source data file). (GCN2iB = grey bars and A92i = pink bars with W = filled bars and KO = chequered) (**b**–**d**). Structural models of GCN2iB and A-92 inhibitors. The Tyr-down position and RS3 space are shown as yellow and blue space filling spheres, respectively. **b** Crystal structure of GCN2 in complex with GCN2iB (PDB code 6N3O). **c**, **d**. Models of GCN2 in complex with A-92, generated using molecular docking against PDB code 6N3O. Two alternative Type I binding poses are shown.

---

manufacturer's instructions, and cells were resuspended in 1 μg/mL DAPI/PBS (Sigma-Aldrich, Dorset, UK). DAPI and EdU staining was assessed with a FACS LSRII (BD Biosciences, Oxford, UK), counting 10000 cells per sample. Gating strategy is shown in supplementary Fig. 2. Data was analyzed using FlowJo software (FlowJo, Oregon, USA) https://www.flowjo.com/solutions/flowjo.

**RNA-seq**

At the Babraham Institute NCI-H358 cells were treated DMSO, 1 μM Dabrafenib (Dab), 30 nM Selumetinib (Sel) or 1 μM Dabrafenib + 30 nM Selumetinib (Dab+Sel) for 4hrs or 24 hrs. Four independent biological repeats were prepared making a total of 32 samples. Cells were lysed in RLT buffer, with the addition of β-mercaptoethanol. Lysates were

disrupted using Qiashredder columns (Qiagen) and RNA prepared as described in the manufacturers protocol (Qiagen, RNeasy plus) with samples eluted in RNase-free water. Samples were analysed by Nano-drop spectrophotometer before being shipped to Plexxikon Inc., CA. Post shipping, Nanodrop analysis was repeated and fragment analysis was performed for quality control purposes at Plexxikon before shipping to the UC Berkeley QB3 Genomics Facility. Strand specific RNA-seq libraries were prepared from poly-A enriched RNA using a Kapa Biosystems Kit and then sequenced (150 bp paired end reads) on a NovaSeq S4 flow cell (QB3 Genomics, UC Berkeley, Berkeley, CA, RRID:SCR_022170). Reads were trimmed for quality and to remove adaptor sequences using fastp version 0.23.4[62]. Trimmed reads were then quantified with salmon (v1.10.1) using the hg38 transcriptome index with full decoy sequences downloaded from RefGenie[63,64]. The R package tximeta was used to calculate offsets, annotate and sum-marize expression to the gene level[65]. Then, the edgeR package was used to filter out genes not expressed in any sample, estimate dis-persion, fit a generalized linear model and identify differentially expressed genes[66]. Finally, the camera function from the limma pack-age was used to test whether sets of genes (from the molecular sig-natures database Hallmark genesets) were highly ranked relative to other genes in terms of differential expression[67]. Data are available at Gene Expression Omnibus (https://www.ncbi.nlm.nih.gov/geo/) under accession number GSE271504.

### Preparation of cell lysates for SDS-PAGE and Western blotting
Culture medium from cells growing on dishes was removed. Cells were washed with PBS and then and harvested in lysis buffer (20 mM Tris [pH 7.5], 137 mM NaCl, 1 mM EGTA, 1% (v/v) Triton X-100, 10% (v/v) glycerol, 1.5 mM MgCl2, 1 mM Na3VO4, 1 mM PMSF, 20 mM leupeptin, 10 mg/ml aprotinin and 50 mM NaF). Cell extracts were snap frozen, thawed and cleared by centrifugation. Supernatant protein con-centration was determined by Bradford protein assay (Bio-Rad) and absorbance measured at 562 nm using a PHERAstar FS plate reader (BMG Labtech, Aylesbury, UK). Samples were prepared for SDS-PAGE by boiling for 5 min in 1 x Laemmli sample buffer (50 mM Tris-HCl (pH 6.8), 2% (w/v) SDS, 10% (v/v) glycerol, 1% (v/v) β-mercaptoethanol, 0.01% (w/v) bromophenol blue).

### SDS-PAGE and western blotting
Lysates were separated by SDS-PAGE (Mighty small II gel apparatus, Hoefer, Massachusetts, USA). Polyacrylamide gels consisted of a resolving phase of 8−16% (w/v) acrylamide (37.5:1 acrylamide:bisacry-lamide, 2.7% crosslinker; Bio-Rad, Watford, UK), 0.375 M Tris-HCl (pH 8.8), 0.2% (w/v) SDS (Bio-Rad, Watford, UK), 0.1% (w/v) ammonium persulfate, 0.1% TEMED (Bio-Rad, Watford, UK) and a stacking phase of 4.5% (w/v) acrylamide (37.5:1 acrylamide:bisacrylamide, 2.7% cross-linker), 0.125 M Tris-HCl (pH 6.8), 0.2% (w/v) SDS (Bio-Rad, Watford, UK), 0.1% ammonium persulfate, 0.125% TEMED. Gels were run using running buffer (0.2 M glycine, 25 mM Tris, 0.1% (w/v) SDS) and a cur-rent of 15 mA per gel for 3-4 hours. Gels were then blotted by wet transfer (Bio-Rad, Watford, UK) to methanol activated PVDF (Immo-bilon-P Membrane, Merck Millipore, Watford, UK) using transfer buffer (0.2 M glycine, 25 mM Tris, 20% (v/v) methanol) and a current of 300 mA for 90-180 min. Membranes were blocked in 5% milk/TBST (5% (w/v) non-fat powdered milk, 10 mM Tris-HCl (pH 8.0), 150 mM NaCl, 0.1% (v/v) Tween-20) for 1 hour at room temperature. Membranes were incubated with primary antibodies as recommended in 5% milk/TBST or 5% BSA/TBST overnight at 4 °C with agitation. Antibodies are detailed in Table S1. Membranes were then washed in TBST for 3 ×10 min and incubated with horseradish peroxidase-conjugated sec-ondary antibodies (Bio-Rad, Watford, UK) diluted 1:3000 in 5% milk/TBST, or fluorescently labelled secondary antibodies diluted 1:15000 (Cell Signaling Technology, NEB, Hitchin, UK), for 1 hour at room temperature. Membranes were again washed for 3 ×10 min in TBST. Detection was performed using Amersham ECL Western Blotting Detection Reagent (Cytiva), X-ray film and Compact X4 film developer (Xograph, Gloucestershire, UK). Quantification of fluorescently labelled membranes was performed using the Odyssey Infrared imaging system (LI-COR, Cambridge, UK) using Image Studio V5.2 (LI-COR Biosciences https://www.licor.com/bio/products/software/image_studio/).

Statistics were performed using GraphPad Prism 8 or higher (GraphPad Software https://www.graphpad.com/scientific-software/prism). All uncropped blots, quantification and statistics can be found in the source data file.

### GCN2 knockout by CRISPR/Cas9-mediated gene editing
Plasmids containing guide RNAs (gRNAs) to *GCN2;* hsEIF2AK4_gui-de1_pSpCas9(BB)-2A-Puro (MP1) and hsEIF2AK4_guide2_pSpCas9(BB)-2A-Puro (MP1) were a kind gift from Dr H Harding, Cambridge University and were subcloned into a pSpCas9(BB)-2A-GFP genome editing vector, which was a gift from Feng Zhang (Addgene plasmid #48138). Cells were transfected with the *GCN2* gRNA containing Cas9 plasmids using jet-Prime (Polyplus Transfection, Illkirch, France). Transfection was mon-itored by GFP expression and single GFP positive/DAPI negative cells were sorted in to 96 well plates using a 100 μm nozzle on a BD FACSARIA III cell sorter (BD Biosciences, Oxford, UK). Clones of interest were identified by Western blot screening for absence of GCN2.

### DNA sequencing to confirm absence or presence of GCN2 mutations
Genomic DNA was extracted from HCT116 (control untransfected) and all WT and KO clones using Qiagen quick extract according to the manufacturer's protocol. Genomic DNA flanking the CRISPR guide binding site was amplified by PCR using OneTaq (NEB, Hitchin, UK) according to the manufacturer's instructions and using the primers indicated in table S1. The products generated were then cloned into the TOPO-TA cloning vector (Thermo Fisher Scientific, Loughborough, UK) following the manufacturer's instructions. The resulting con-structs were used to transform chemically competent DH5α (NEB, Hitchin, UK), 4-5 of the resulting clones derived from DNA for each cell line were sent for sequencing (Genewiz, Bishop's Stortford, UK).

### GCN2 kinase assays
Expression and purification of recombinant human GCN2 and eIF2α was conducted as described previously[68]. 10 nM GCN2 was incubated at 30 °C for 20 min with 2 μM eIF2α and 10 μM ATP. Samples were then quenched via the addition of SDS-PAGE loading buffer and heating to 95 °C for 3 min. Samples were separated on by SDS-PAGE (5-12% Bis-Tris Gel), run at 180 V for 35 min in MES buffer (50 mM MES (2-[N-morpholino]ethanesulfonic acid), 50 mM Tris base, 1 mM EDTA. 0.1% (w/v) SDS). Gels were then briefly incubated in 20% ethanol solution prior to a transfer to a nitrocellulose membrane using the iBlot 2 Transfer system (ThermoFisher Scientific). Membranes were pro-cessed as above except LICOR secondary antibodies were used.

### Modelling of GCN2-inhibitor complexes
GCN2-ligand complex models were generated using the Webina implementation of AutoDock Vina[69] based on relevant crystal struc-tures (Dabrafenib and GDC0879 used GCN2 bound to a Type I.5 inhi-bitor, PDB code 6N30[38]; LY3002190 used GCN2 bound to a Type II inhibitor, PDB code 6N3L[38]). Docking poses were selected based on closest match to the equivalent BRAF crystal structures. Met802Phe structure was modelled in PyMOL, based on the AlphaFold2 model of active GCN2, taken from KinCore database[70,71]. The structure of inac-tive, yeast GCN2 was used to model the position of Tyr651 in the Tyr-down position[42] (PMID: 15964839).

## Transfection

Cells were cultured to approximately 60% confluency. Constructs were then transfected using JetPrime polyplus transfection reagent (Polyplus Transfection, Illkirch, France) or lipofectamine 3000 (Thermo-Fisher Scientific) according to manufacturer's instructions. siRNA transfections were performed using RNAiMAX (ThermoFisher Scientific) according to manufacturer's protocols and using 10 nM final siRNA concentration.

## GCN2 expression constructs

GCN2 was amplified from GST-GCN2 (MRC, Dundee PPU-DU67392-proteins-703081) with primers to introduce Xho1 and Xba1 restriction sites at the 5' and 3' of the coding regions respectively. The PCR product was cloned into EGFP-C3 (Clontech). Primers for mutagenesis were designed using NEBaseChanger (sequences shown in Table S1) and mutagenesis was performed using the Q5 Site-Directed Mutagenesis Kit (New England Biolaboratories). DNA was transformed into stable competent cells (New England Biolaboratories). Whole plasmids were fully sequenced using Plasmidsaurus.

## Reporting summary

Further information on research design is available in the Nature Portfolio Reporting Summary linked to this article.

## Data availability

RNA seq data are available at Gene Expression Omnibus under accession number GSE271504. All HCM processed data is shown in the source data file. Raw data is available on request from the corresponding author. All uncropped blots and any quantification and statistical analysis of western blot data is shown in the source data file. Graphpad files are available upon request. Data from FLOW analysis in supplementary Fig. 2 is provided in the source data file. PDB code 6N30 PDB code 6N3L PDB code 1ZY4 (structure of inactive yeast GCN2) [https://doi.org/10.2210/pdb1ZY4/pdb]. Source data are provided with this paper.

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

## Acknowledgements

We would like to thank past and present members of the Cook lab for their support and encouragement. We thank Rahul Samant, Ian McGough and Hayley Sharpe (Signalling Programme, the Babraham Institute) and Benedict Cross (PhoreMost) and Paul Smith for suggestions and encouragement. We would like to acknowledge the excellent support of the Babraham Institute Science Services including Simon Walker and Hanneke Okkenhaug (Imaging), Rachael Walker (Flow Cytometry) and Hayley Carr (Statistical analysis) supported by a Core Capability Grant from the Biotechnology and Biological Sciences Research Council (UKRI-BBSRC). Work in Simon Cook's laboratory was supported by Plexxikon Inc (AMK and SJC) and Institute Strategic Programme Grants BB/J004456/1, BB/P013384/1 and BB/Y006925/1 from

UKRI-BBSRC (SJC and RG). Work in Glenn Masson's laboratory was supported by a University of Dundee Baxter Fellowship and a TENOVUS Grant T21/01 119615. Work in Patrick Eyer's laboratory was supported by UKRI-BBSRC grants BB/S018514/1 and BB/X002780/1 while work in Richard Bayliss's laboratory was supported by Cancer Research UK (C24461/A23302).

## Author contributions

Rebecca Gilley (R.G.): Conceptualisation, Investigation, Formal analysis, Writing—review and editing. Andrew Kidger (A.K.): Conceptualisation, Investigation, Formal analysis, Writing—review and editing. Graham Neill (G.N.): Conceptualisation, Investigation, Formal analysis. Eve Morrison (E.M.): Investigation. Paul Severson (P.S.): Formal analysis. Dominic Byrne (D.B.): Investigation. Niall Kenneth (N.K.): Investigation, Writing—review and editing. Gideon Bollag (G.B.): Resources, Writing—review, Funding acquisition and editing. Chao Zhang (C.Z.): Resources, Writing—review, Funding acquisition. Taiana Maia de Oliveira (T.M.O.): Conceptualisation, Writing—review and editing. Patrick Eyers (P.E.): Conceptualisation, Writing—review and editing; Richard Bayliss (R.B.): Conceptualisation, Writing—review and editing. Glenn Masson (G.M.): Investigation, Resources, Supervision, Funding acquisition, Writing—review and editing. Simon Cook (S.J.C.): Conceptualisation, Resources, Supervision, Funding acquisition, Writing—original draft, review and editing, Project administration. S.C., R.G. and A.K. conceived the study. A.K., R.G., and S.C. designed experiments. A.K. performed HCM and biochemical experiments. R.G. performed molecular biology and biochemical experiments. G.N. and G.M. performed all in vitro kinase assays. N.K. analysed K.O. MEFs. RB built and refined structural models. S.C. wrote the draft manuscript. All authors reviewed and edited the manuscript.

## Competing interests

PS, GB and CZ were paid employees of Plexxikon Inc. TMO is a paid employee of AstraZeneca. The initial stages of this work were supported by a sponsored research collaboration in SJC's laboratory funded by Plexxikon Inc; this paid AK's salary; SJC received no remuneration from Plexxikon. The remaining authors declare no competing interests.

## Additional information

[1]Signalling Programme, The Babraham Institute, Babraham Research Campus, Cambridge CB22 3AT, UK. [2]Division of Cellular and Systems Medicine, School of Medicine, University of Dundee, Dundee, UK. [3]Plexxikon Inc., 1 Bolivar Drive, Berkeley, CA 94710, USA. [4]Department of Biochemistry, Cell and Systems Biology, Institute of Systems, Molecular & Integrative Biology, University of Liverpool, Crown Street, Liverpool L69 7ZB, UK. [5]Mechanistic and Structural Biology, Discovery Sciences, BioPharmaceuticals R&D, AstraZeneca, Cambridge, UK. [6]Astbury Centre for Structural Molecular Biology, School of Chemistry, University of Leeds, Leeds, UK. [7]Present address: Andrew Kidger, Early Oncology R&D, AstraZeneca, Cambridge, UK. [8]Present address: Paul Severson, Tupos Therapeutics, Inc., 25801 Industrial Blvd, Ste 100, Hayward, CA 94545, USA. [9]Present address: Chao Zhang, Tupos Therapeutics, Inc., 25801 Industrial Blvd Ste 100, Hayward, CA 94545, USA. [10]Present address: Gideon Bollag, Opna Bio LLC, 600 Gateway Blvd, Ste 100, South San Francisco, CA 94080, USA. ✉e-mail: becky.gilley@babraham.ac.uk; simon.cook@babraham.ac.uk

