## [Transparent Peer Review file · Nature Communications]

RAF inhibitors activate the integrated stress response by direct activation of GCN2

Corresponding Author: Dr Simon Cook

Version 0:

Reviewer comments:

Reviewer #1

(Remarks to the Author)

This manuscript demonstrates that multiple Raf kinase inhibitors, including the clinically approved vemurafenib, dabrafenib, and encorafenib, activate the integrated stress response (ISR) through a mechanism dependent on GCN2 catalytic activity. The findings build upon two recent studies (Tang et al., 2022 and Szaruga et al., 2023) which showed GCN2-dependent ISR activation by EGFR and PERK/PKR inhibitors. Notably, the Raf inhibitors trigger ISR activation at therapeutically relevant concentrations, suggesting that this effect should be considered when evaluating the clinical efficacy of these compounds.

Overall, this manuscript represents a significant body of work. The authors convincingly demonstrate that several Raf inhibitors lead to ERK activation-independent inhibition of DNA synthesis (Fig. 1 and Sup. Fig. 1). RNA-seq experiments designed to identify ERK activation-independent gene sets modulated by dabrafenib (Fig. 2) show that the UPR is activated. Follow-up experiments (Fig. 3) convincingly demonstrate a rapid increase in ATF4 expression, suggesting activation of the integrated stress response. The authors then show that multiple Raf inhibitors increase p-T899 in GCN2 and p-S51 eIF2 α , although these effects are both fairly modest (as described below). They also demonstrate that ISRIB and another ATP-competitive GCN2 inhibitor (A-92) block increases in ATF4 and CHOP expression (Fig. 4b,c). Multiple GCN2 knockdown and knockout experiments (Fig. 5) convincingly show that GCN2 is required for BRAf inhibitor-promoted increases in ATF4 and CHOP expression. Together, this manuscript provides convincing evidence that several BRAf inhibitors lead to increases in multiple markers of the integrated stress response.

While the aforementioned results are convincing, the evidence demonstrating that Raf inhibitors activate the ISR through direct engagement of GCN2's ATP-binding site is significantly less conclusive. The most direct evidence of ATP-binding site engagement is the A-92 competition experiments in Fig. 4C and the experiments with drug-resistant gatekeeper mutants in Fig. 9. However, for the A-92 competition experiments, only reductions in ATF4 and CHOP expression are probed.

-To provide a more direct readout of effects on GCN2, the ability of A-92 to prevent BRAf inhibitor-promoted formation of p-T899 GCN2 should be probed

Relatedly, the authors use GDC-0879 as a control compound (and the basis for their structural modeling of why certain inhibitors lead to GCN2 activation) that does not activate GCN2 and the ISR. However, is there any evidence that this inhibitor occupies the ATP-binding site of GCN2? The experiments in Figure 7B suggest that it does not, as even very high concentrations (>50 μ M) do not lead to a diminution of p-GCN2.

-Again, A-92 seems a more appropriate compound for in vitro assays and structural modeling. An in vitro competition experiment demonstrating that A-92 blocks the Raf inhibitor-promoted increases in p-T899 GCN2 that is observed in Fig. 7A would provide more confidence that Raf inhibitors are modulating GCN2 through its ATP-binding site.

The additional data suggesting that Raf inhibitors are modulating GCN2 through the ATP-binding site of its kinase domain are the cellular results showing Raf inhibitors cannot activate drug-resistant gatekeeper mutants (why smaller gatekeeper substitutions are drug-resistant is a little unclear to me). However, I could not find evidence that these mutants can be activated like wild-type GCN2.

-Given that mutations at this site (the M802F) can constitutively activate GCN2 and the ISR, evidence that Ala and Gly gatekeeper mutants can be activated (e.g., by treatment with histidinol) should be provided to rule out the possibility that these are just kinase-dead mutants.

he most parsimonious explanation for how BRAf inhibitors promote GCN2's activation is through engagement of the kinase domain's ATP-binding site, and I find this mechanism likely. However, given that so many structurally different inhibitors (EGFR, PERK, Raf) promote activation, more conclusive evidence is needed.

Regarding the model of how BRAf inhibitors activate GCN2, several mechanistic details remain unclear. The authors draw analogies to the paradoxical activation of wild-type Raf kinases. Do they propose that Raf inhibitors also promote GCN2 dimerization despite what appears to be fairly high levels of dimerization? This seems like a testable hypothesis. Alternatively, are the authors proposing that BRAf inhibitors allosterically activate one protomer of pre-formed GCN2 dimers? If it is the latter, I cannot think of any results that rule out an alternative model in which Raf inhibitor-bound GCN2 is more efficiently phosphorylated by uninhibited GCN2. Although I could not fully understand the logic behind the experiment performed for Fig. 5D in the accompanying paper (Neill et al.), it appears that encorafenib binding does make T899 of GCN2 a better substrate for uninhibited GCN2, which supports the alternative model. The authors point to the bell-shaped curve for p-T899 GCN2 in the presence of Raf inhibitors, but I think this would also result from the alternative model described above. Demonstrating a more robust effect on GCN2's substrate pS51 eIF2 α in *in vitro* assays would provide more convincing evidence for the authors' model. Again, the accompanying paper provides data (a decrease in p-S51, labeled as p-S52, in the presence of multiple Raf inhibitors (Fig. 1C) that contradicts an allosteric activation model that (I think) is proposed in this paper.

There are also several other data analysis issues and/or inconsistencies that should be addressed:

- The overall effects of BRAf inhibitors on p-T899 GCN2 levels are fairly subtle, and in several cases (e.g., Fig. 5A, 5D, 6C), overall GCN2 levels change, making it difficult to assess how consistently these effects are observed. The authors state that the western blots shown in the figures are representative of three biological replicates. They should use these replicates to quantify p-T899 GCN2 (normalized to total GCN2) levels and perform statistical analyses to compare DMSO to Raf inhibitor-treated cells.

- Overall effects on p-S51 eIF2 α are even more subtle. Again, replicates should be quantified, and statistical comparisons should be performed. The authors acknowledge how weak the effects on p-S51 eIF2 α are and suggest that this is due to inhibitor binding blocking substrate binding, but wouldn't eIF2 α be phosphorylated at the uninhibited protomer?

- The dose responses in Figure 6C don't support the authors' model, as there is a lack of correlation between p-T899 GCN2 levels and ATF4 and CHOP expression. It also appears that control compound GDC-0879 leads to an increase in ATF4 expression, although to a lesser extent than the other inhibitors.

- It is difficult to interpret the HT29 data in Figure 6B. There is perhaps a slight increase in p-T899 GCN2 levels, but GCN2 levels increase as a whole.

- The authors point to bell-shaped curves for the cellular effects of Raf inhibitors. However, in several cases, these statements are not supported by the data provided (e.g., a bell-shaped curve is not observed for p-T899 GCN2 for any of the inhibitors and for ATF4 expression for encorafenib in Fig. 6C).

- It is confusing that A-92 increases p-T899 GCN2 levels in Fig. 10A. Additionally, why were p-S51 levels not probed in 10A?"

Minor:

- Figure 7. What is +ve ATP in the first lane?

Reviewer #2

(Remarks to the Author)

Although targeting RAF kinase, particularly BRAF(V600E) mutant with RAFi for treating cancers has achieved promising outcome, drug resistance and toxicity remains a major challenge in current clinic practice. Understanding underlying mechanisms has important implications for both cancer research and treatment. In this manuscript entitled "RAF inhibitors activate the integrated stress response by direct activation of GCN2", Gilley et al. found that RAFi induced ERK-independent/eIF2 α -dependent expression of ATF4 and CHOP through activating unfolded protein response. Further, they showed that GCN2 was a key factor for RAFi-induced ATF4 and CHOP expression by using GCN2 inhibitor, GCN2 RNAi, and GCN2 KO system. More importantly, using biochemical, molecular modelling and mutagenesis approaches, they demonstrated that RAFi directly associated with GCN2 and enhanced GCN2 dimerization, which leads to the paradoxical activation of GCN2 as RAFi does on RAF kinase. Overall, this is a very interesting study with solid biochemical data, and somehow contributes targeted cancer therapy with RAF inhibitors, since RAFi-induced ISR(integrated stress response) may provide cancer cells a time window for developing acquired resistance. Although it has been reported by other groups that RAFi activates GCN2-eIF2 α -ATF4 axis and hence induces ISR(i.e. Nagasawa et al., BBRC, 2017; Nagasawa et al., iScience, 2020), it is novel that authors identified GCN2 kinase as the direct target of RAFi and GCN2 was transactivated by RAFi. However, there are some significant issues needed to be addressed in this manuscript before considering for publishing.

Major point:

1. The most important finding of this study is that RAFi binds to and transactivates GCN2 through inducing dimerization as it does on RAF. However, authors' data support this finding is not solid. To strengthen this point, firstly they need to provide the data of RAFi affinity with GCN2, since they already purified GCN2 protein and could get this data easily by using Surface Plasmon Resonance (SPR) or Microscale Thermophoresis (MST) approaches. Secondly, for their molecular modelling of RAFi-GCN2 complex, it should be validated by using mutagenesis approach since it's just a prediction that needs to confirm by using other approach. In addition, the RAFi-driven transactivation of GCN2 is also needed to confirm by mutating the GCN2 dimer interface.

2. Some experiments (i.e. experiments in Fig1~4) were carried out on single cell line, NCI-H358. To exclude the cell line-specific effect, those data should be validated by using at least one more cell line. Since RAF inhibitors have been applied

to treat BRAF(V600E)-mutated melanoma and other cancers, we suggest, authors may use a BRAF(V600E)-harboring melanoma cell line to validate those data, which could also elevate the clinic significance of their findings.

Minor point:

1. Line 55-57, "BRAFFV600E/K.....signal as constitutively active monomers." The monomer hypothesis of BRAF(V600E) was raised up for explaining the RAFi resistance before, but so far there's no solid data supporting this opinion. On the other sides, several groups had provided strong evidence that RAF kinases as well as their mutants including BRAF(V600E) function as dimers or oligomers to activate MEK (Diedrich et al., EMBO J, 2017; Yuan et al., oncogene, 2018; Cope et al., Chembiochem, 2019; Lauinger et al., Sci Adv, 2023). This should be corrected.
2. Figure 5c, the quality of phospho-GCN2 blot is low, please repeat this blot and get a better image. Also, samples for WT, GCN2 KO should be run on one blot, otherwise, they are not comparable. This issue also existed in 5D.
3. In Figure 5c-d, Figure 6b-c, Figure 9a-c, and Figure 10a, is HSP90 used a loading control or ISR marker?
4. Figure 7c, it's better to re-organize the sample orders, samples treated with a same drug at different concentrations should be put together, which will be more convenient for comparing drugs' effect.
5. Figure 9a, the GFP-GCN2 blot needs to be improved.
6. Figure 10a, it's better to run all samples on one blot.
7. Reference#39, it's a repetitive citation. Please delete it.

Reviewer #3

(Remarks to the Author)

The manuscript by Gilley et al. describes studies of RAF inhibitors prompted by observations of reduced DNA replication that the authors use to zero in on an unexpected activation of the ISR. Using genetic and biochemical approaches the authors show that interactions with GCN2 are important to activate the ISR and block DNA replication in the RAFi treated cells.

Overall, the work was thorough and methodical and I had very few concerns. Below are some queries and suggestions.

- In Figure 2, BiP seems to be induced by Selumetinib after 24 hours in panel A, but it is not induced in panel B. Why? In general the authors rely on western blots where the differences are usually quite stark. However, in some instances that are more subtle quantification would be helpful.

- In Figure 5, the authors claim that GCN2iB causes an increase in ATF4 and CHOP. While the increase in ATF is clear enough, I don't see a significant increase in CHOP. In other figures this induction is time sensitive and lags after ATF4 so perhaps it is a timing issue. Regardless, can the authors provide stronger support for this statement?

- There are a large number of inhibitors and signaling outputs tested which can be somewhat confusing. I suggest the authors draw a model figure at the end summing up the major findings of the work. Highlighting, in particular, the potential clinical impacts of their work.

Reviewer #4

(Remarks to the Author)

Version 1:

Reviewer comments:

Reviewer #1

(Remarks to the Author)

The authors have adequately addressed all of the experimental concerns raised during the previous round of review. I still feel that the use of GDC-0879 for structural modeling "In contrast, GDC-0879, a Type I inhibitor, does not occupy the RS3 space (Fig 8E)." is inappropriate since the authors state that they have no evidence that this inhibitor binds to GCN2.

Reviewer #2

(Remarks to the Author)

Authors have addressed most of my questions. I do not have no further comments.

Reviewer #3

(Remarks to the Author)

The authors provided a thorough rebuttal with important new data and revisions. I have no additional concerns.

Reviewer #4

(Remarks to the Author)

RAF inhibitors activate the Integrated Stress Response by direct activation of GCN2

Reviewer #1 (Remarks to the Author):

This manuscript demonstrates that multiple Raf kinase inhibitors, including the clinically approved vemurafenib, dabrafenib, and encorafenib, activate the integrated stress response (ISR) through a mechanism dependent on GCN2 catalytic activity. The findings build upon two recent studies (Tang et al., 2022 and Szaruga et al., 2023) which showed GCN2-dependent ISR activation by EGFR and PERK/PKR inhibitors. Notably, the Raf inhibitors trigger ISR activation at therapeutically relevant concentrations, suggesting that this effect should be considered when evaluating the clinical efficacy of these compounds.

Overall, this manuscript represents a significant body of work. The authors convincingly demonstrate that several Raf inhibitors lead to ERK activation-independent inhibition of DNA synthesis (Fig. 1 and Sup. Fig. 1). RNA-seq experiments designed to identify ERK activation-independent gene sets modulated by dabrafenib (Fig. 2) show that the UPR is activated. Follow-up experiments (Fig. 3) convincingly demonstrate a rapid increase in ATF4 expression, suggesting activation of the integrated stress response. The authors then show that multiple Raf inhibitors increase p-T899 in GCN2 and p-S51 eIF2 α , although these effects are both fairly modest (as described below). They also demonstrate that ISRIB and another ATP-competitive GCN2 inhibitor (A-92) block increases in ATF4 and CHOP expression (Fig. 4b,c). Multiple GCN2 knockdown and knockout experiments (Fig. 5) convincingly show that GCN2 is required for BRAf inhibitor-promoted increases in ATF4 and CHOP expression. Together, this manuscript provides convincing evidence that several BRAf inhibitors lead to increases in multiple markers of the integrated stress response.

Response. We thank the referee for their kind comments and the useful queries raised below.

1. While the aforementioned results are convincing, the evidence demonstrating that Raf inhibitors activate the ISR through direct engagement of GCN2's ATP-binding site is significantly less conclusive. The most direct evidence of ATP-binding site engagement is the A-92 competition experiments in Fig. 4C and the experiments with drug-resistant gatekeeper mutants in Fig. 9. However, for the A-92 competition experiments, only reductions in ATF4 and CHOP expression are probed. To provide a more direct readout of effects on GCN2, the ability of A-92 to prevent BRAf inhibitor-promoted formation of p-T899 GCN2 should be probed.

Response. Please see revised figure Supp 5A and Main Figure 9B. In Supp Figure 5A we have quantified the effects of Dabrafenib on p-GCN2 (normalising to total GCN2) from multiple independent experiments. We show that Dabrafenib drives a statistically significant increase in p-GCN2 that is reversed by A-92 – consistent with it competing with Dabrafenib at the ATP-binding site of GCN2. In contrast, and as a control, ISRIB does not reverse the increase in p-GCN2 as it acts downstream of GCN2 by antagonising p-eIF2 \$\alpha\$. In the new Figure 9B we have undertaken a more detailed concentration-response analysis of the 802A gatekeeper mutation, comparing it with WT when re-expressed in

GCN2 KO HCT116 cells. Re-expression of WT GCN2 reconstitutes the Dabrafenib-induced increase in p-GCN2, which is maximal at 1 μ M, consistent with our previous results. In contrast, expression of GCN2 M802A (with a small Ala residue) in the GCN2 KO cells provides allows easier access to Dabrafenib, conceptually identical to early work with p38 MAPK's gatekeeper residue where Thr residue (p38 α/β) or Met residue (p38 γ/δ) mutated to Ala acting as a 2-4log sensitizer to pyridinyl imidazoles (Refs 47 and 48). This is manifest as a pronounced left shift in the concentration response curve for Dabrafenib induced p-GCN2 and CHOP expression, now maximal at 1-3 nM, compared to re-expression of WT GCN2.

This new data in Supp 5B and Main Figure 9B, along with our previous data focused on ATF4 and CHOP (Supp Figure 5A and Main Figure 4C) is complemented by the companion paper from our collaborator Glenn Masson (<https://doi.org/10.1101/2024.08.14.606984>) which has deployed HDX-MS to demonstrate that Dabrafenib, Encorafenib and GCN2iB all bind within the kinase domain of GCN2 (see point 5 below).

Together this data unambiguously demonstrates that Dabrafenib directly engages the ATP binding site of the canonical kinase protomer in GCN2, which is consistent with the structural models presented in Figure 8.

2. Relatedly, the authors use GDC-0879 as a control compound (and the basis for their structural modeling of why certain inhibitors lead to GCN2 activation) that does not activate GCN2 and the ISR. However, is there any evidence that this inhibitor occupies the ATP-binding site of GCN2? The experiments in Figure 7B suggest that it does not, as even very high concentrations (>50 μ M) do not lead to a diminution of p-GCN2.

Response. We agree with the referee completely. We have absolutely no data that suggests GDC-0879 binds to the ATP-binding site and activates GCN2; for these reasons we have used it as a negative control throughout our paper.

3. Again, A-92 seems a more appropriate compound for in vitro assays and structural modeling. An in vitro competition experiment demonstrating that A-92 blocks the Raf inhibitor-promoted increases in p-T899 GCN2 that is observed in Fig. 7A would provide more confidence that Raf inhibitors are modulating GCN2 through its ATP-binding site.

Response. As above (addressing point 1), our cellular A-92 competition data and gatekeeper mutations together with the HDX-MS data from our collaborators now confirm beyond all doubt that RAF inhibitors are modulating GCN2 through its canonical ATP-binding site.

4. The additional data suggesting that Raf inhibitors are modulating GCN2 through the ATP-binding site of its kinase domain are the cellular results showing Raf inhibitors cannot activate drug-resistant gatekeeper mutants (why smaller gatekeeper substitutions are drug-resistant is a little unclear to me). However, I could not find evidence that these mutants can be activated like wild-type GCN2.

- Given that mutations at this site (the M802F) can constitutively activate GCN2 and the ISR, evidence that Ala and Gly gatekeeper mutants can be activated (e.g., by treatment with histidinol) should be provided to rule out the possibility that these are just kinase-dead mutants.

Response. To test the functionality of the GCN2 gatekeeper mutants we have employed NXP800 an oral, small molecule inhibitor of the HSF1 pathway (Pasqua AE et.al., J Med

Chem. 2023 Apr 27;66(8):5907-5936) being developed by Nuvectis. NXP800 serves as a very effective indirect activator of GCN2. In new Figure 9A we compared re-expression of WT GCN2 with re-expression of the M802A mutant. For these experiments we employed 1 μ M Dabrafenib as this is optimal for activation of WT GCN2. As described in point 1 above and new Figure 9B, the M802A mutant is left-shifted (sensitized) to Dabrafenib so it is activated at 1-3 nM and inhibited by 1 μ M Dabrafenib. The results clearly show that the gatekeeper Ala mutant is activated strongly by NXP800, demonstrating that this mutant is functional and can be activated. Finally, the re-expressed kinase dead K619A mutant is non-functional and does not respond to either Dabrafenib or NXP800. Together these results confirm that the M802A mutant is a functional kinase, validating the 'left shift' in sensitivity (new Figure 9B) to Dabrafenib as direct evidence for binding of Dabrafenib to the canonical ATP-binding site of GCN2.

The most parsimonious explanation for how BRAf inhibitors promote GCN2's activation is through engagement of the kinase domain's ATP-binding site, and I find this mechanism likely. However, given that so many structurally different inhibitors (EGFR, PERK, Raf) promote activation, more conclusive evidence is needed.

Response. The preceding data, described above in Points 1-4, now conclusively show that Dabrafenib promotes GCN2 activation by binding directly to the kinase domain's canonical ATP binding site and this is supported by our *in silico* structural models and the HDX-MS data of our collaborator Glenn Masson. See Point 5 below also.

(<https://doi.org/10.1101/2024.08.14.606984>).

5. Regarding the model of how BRAf inhibitors activate GCN2, several mechanistic details remain unclear. The authors draw analogies to the paradoxical activation of wild-type Raf kinases. Do they propose that Raf inhibitors also promote GCN2 dimerization despite what appears to be fairly high levels of dimerization? This seems like a testable hypothesis. Alternatively, are the authors proposing that BRAf inhibitors allosterically activate one protomer of pre-formed GCN2 dimers? If it is the latter, I cannot think of any results that rule out an alternative model in which Raf inhibitor-bound GCN2 is more efficiently phosphorylated by uninhibited GCN2. Although I could not fully understand the logic behind the experiment performed for Fig. 5D in the accompanying paper (Neill et al.), it appears that encorafenib binding does make T899 of GCN2 a better substrate for uninhibited GCN2, which supports the alternative model. The authors point to the bell-shaped curve for p-T899 GCN2 in the presence of Raf inhibitors, but I think this would also result from the alternative model described above.

Response. To be clear, we have no evidence and do not believe that RAFi promote dimerization of GCN2. Reports have shown that GCN2 is an obligate dimer. Rather, our results are consistent with low concentrations of RAFi binding to one protomer of a pre-formed dimer, and allosterically-promoting the unbound dimer partner, which is free to engage in ATP-dependent catalysis.

However, the referee is quite right about potential models. We envisage that at low doses RAFi bind to the ATP binding site of one protomer in a pre-existing GCN2 dimer and inhibit it but also promote allosteric changes in the drug-free dimer partner, resulting in its activation, presumably through alignment of hydrophobic spines and repositioning of the helix α C.

Evidence supporting this includes:

(i) the ability of A-92, an ATP competitive GCN2 inhibitor to antagonise RAFi-induced GCN2-ISR activation;

(ii) our molecular modelling is consistent with Dabrafenib, LY3009120 or GCN2iB binding and promoting the assembly of the R-spine whereas GDC-0879 is not predicted to assemble the R-spine and does not activate GCN2 in vitro or in cells. Similarly, A-92 is not predicted to assemble the R-spine and does not activate GCN2 in cells but rather inhibits it;

(iii) The detailed SPR and MST assays proposed are beyond the scope of our cell-based study. However, the parallel paper (<https://doi.org/10.1101/2024.08.14.606984>) from our collaborator Glenn Masson focuses on biophysical approaches to provide critical further molecular and structural insights. First, Thermal Shift Assays show that Dabrafenib, Encorafenib and GCN2iB all produce significant shifts in stability of purified full length GCN2 dimers, indicative of binding, while GDC-0879 produced no shift. Second, HDX-MS analysis shows that Dabrafenib, Encorafenib and GCN2iB all bind within the kinase domain of GCN2, causing similar large reductions in solvent exchange in peptides surrounding the ATP-binding pocket (residues 587-592 (P-Loop), 803-818 (Activation Loop), 836-876 (DFG Loop), 902-914 (T904 auto-phosphorylation site)).

Whether RAFi-induced GCN2 phosphorylation is “cis-autophosphorylation” where one GCN2 kinase phosphorylates its own T899 residue or “trans-autophosphorylation” with the T899 residue of the dimer partner being phosphorylated remains to be resolved. T899 phosphorylation of the drug-bound dimer partner by the drug-free allosterically activated partner seems more plausible to us. A variation on this model is presented in the companion paper by Glenn Masson (<https://doi.org/10.1101/2024.08.14.606984>) in which he demonstrates that in the test tube at least RAFi binding to one GCN2 dimer may make it a better substrate for phosphorylation by another GCN2 dimer. Whether dimer-to-dimer phosphorylation occurs in cells remains to be seen.

Regardless of these variations on the model, subsequent dissociation of RAFi (by competition with ATP) from ‘poised, phosphorylated’ GCN2 would then allow it to phosphorylate eIF2 α . This is consistent with the new *in vitro* GCN2 kinase data (Figure 7C); in our previous data very little RAFi-driven phosphorylation of eIF2 α was observed at 10 μ M ATP, though this supported robust GCN2 T899 phosphorylation. p-eIF2 α is now clearly observed at 100 μ M ATP, consistent with ATP competing RAFi off phosphorylated GCN2, essentially removing inhibition and allowing it to phosphorylate its substrate. We now describe these various scenarios in the discussion.

Demonstrating a more robust effect on GCN2's substrate pS51 eIF2 α in *in vitro* assays would provide more convincing evidence for the authors' model. Again, the accompanying paper provides data (a decrease in p-S51, labeled as p-S52, in the presence of multiple Raf inhibitors (Fig. 1C) that contradicts an allosteric activation model that (I think) is proposed in this paper.

Response. We have addressed the *in vitro* anomaly of p-S51 phosphorylation as described above. Binding of the drug to the ATP pocket represses the kinase *in vitro* and also prevents substrate binding. Raising ATP levels to 100 μ M (compared with our previous 1 μ M) competes RAFi off, allowing much stronger phosphorylation of S51 of eIF2 α , whereas GCN2 autophosphorylation at T899 was readily detectable at 1 μ M ATP.

6. There are also several other data analysis issues and/or inconsistencies that should be addressed:

A. The overall effects of BRAF inhibitors on p-T899 GCN2 levels are fairly subtle, and in several cases (e.g., Fig. 5A, 5D, 6C), overall GCN2 levels change, making it difficult to assess how consistently these effects are observed. The authors state that the western blots shown in the figures are representative of three biological replicates. They should use these replicates to quantify p-T899 GCN2 (normalized to total GCN2) levels and perform statistical analyses to compare DMSO to Raf inhibitor-treated cells.

Response. We have addressed this comment in Figure 5 and Figure 6 performing both single time point and timecourse experiments. Our results show that an increase in p-GCN2 (T899) is apparent within 15 mins and rises to a maximum of approx. 3 fold within 4 hours and this is statistically significant. In these experiments we have compared Dabrafenib with other RAFi; Dabrafenib, Encorafenib and LY-3009120 all increase p-GCN2 in a statistically significant manner, whilst GDC-0879 does not, consistent with our wider data set showing that GDC-0879 is not able to activate GCN2. We also benchmarked this data against NXP800 and Histidinol (Figure 5C). In Figure 6, we have benchmarked Dabrafenib against Tunicamycin, an ER stressor; Dabrafenib again activates GCN2 whereas Tunicamycin does not, providing a specificity control.

B. Overall effects on p-S51 eIF2 α are even more subtle. Again, replicates should be quantified, and statistical comparisons should be performed. The authors acknowledge how weak the effects on p-S51 eIF2 α are and suggest that this is due to inhibitor binding blocking substrate binding, but wouldn't eIF2 α be phosphorylated at the uninhibited promoter?

Response. We have now quantified the increase in p-S51 eIF2 α by RAFi from multiple experiments and show (Figures 5 and 6) that, even though it is a modest 1.3-1.5 fold increase, this is statistically significant for Dabrafenib, Encorafenib and LY3009120, but not GDC-0879. It is also a matter of timing too. In Figure 6 there is a modest increase (1.6 to 1.8 fold) from 30 mins to 2 hrs with Dabrafenib treatment but this drops at 4 hrs to 1.2 fold. For single timepoint comparisons (in Figure 5) we typically treat cells for 4 hours with RAFi in order to capture the transcriptional increases in both ATF4 and CHOP levels too.

It is important to recognise that detecting increases in p-S51 eIF2 α is a common challenge in the ISR field as reviewed by Anne Bertolotti and co-workers, experts in the field. Please see "An Overview of Methods for Detecting eIF2 α Phosphorylation and the Integrated Stress Response". In: Matějů, D., Chao, J.A. (eds) *The Integrated Stress Response*. (2022) *Methods in Molecular Biology*, vol 2428. Humana, New York, NY. (https://doi.org/10.1007/978-1-0716-1975-9_1). This comprehensive methods chapter states "phosphorylation of eIF2 α remains difficult to detect and quantify, because of its transient nature and because sub-stoichiometric amounts of this modification are sufficient to profoundly reshape cellular physiology". This review also presents data showing a 2-fold increase in p-S51 eIF2 α with Tunicamycin (an ER stressor and PERK activator) as being a 'typical' response. Indeed, in our experiments Tunicamycin did indeed drive a 1.8 to 2-fold increase in p-S51 eIF2 α but this 'modest' signal is amplified down the pathway resulting in substantial increases in both ATF4 and CHOP expression. Thus, our results with RAFi are significant and benchmarked against a recognised standard in the field.

C. The dose responses in Figure 6C don't support the authors' model, as there is a lack of correlation between p-T899 GCN2 levels and ATF4 and CHOP expression. It also appears that control compound GDC-0879 leads to an increase in ATF4 expression, although to a lesser extent than the other inhibitors.

Response. ATF4 expression is by far the most dynamic marker for activation of the ISR. This has been quantified and shows that Dabrafenib and LY-300 show a bell-shaped curve for ATF4 expression. Encorafenib activates GCN2 at slightly higher concentrations so we don't quite capture the full downward shift of the curve for ATF4 expression. GDC-0879 does not activate GCN2 as judged by no increase in p-GCN2 or ATF4 expression. Dabrafenib, Encorafenib and LY-300 clearly drive GCN2 phosphorylation but this does not exhibit a clear down turn at higher doses like ATF4. There are likely two possible explanations for this. First, this may reflect the time taken for activation (or expression) of phosphatases responsible for dephosphorylating T899. Second, inhibitory feedback mechanisms that shut down GCN2 activity may target sites (or mechanisms) other than T899 in GCN2 to inhibit its activity.

There may be a modest effect of 10 μ M GDC-0879 on ATF4 in that blot panel but (a) this is not a relevant dose for this drug and (b) analysis of replicate samples in the graph accompanying Fig 6C indicates this is not reproducible.

D. It is difficult to interpret the HT29 data in Figure 6B. There is perhaps a slight increase in p-T899 GCN2 levels, but GCN2 levels increase as a whole.

Response. This has now been quantified from multiple experiments and is statistically significant for both HCT116 and HT29 cells.

E. The authors point to bell-shaped curves for the cellular effects of Raf inhibitors. However, in several cases, these statements are not supported by the data provided (e.g., a bell-shaped curve is not observed for p-T899 GCN2 for any of the inhibitors and for ATF4 expression for encorafenib in Fig. 6C).

Response. Please see response to 6C above, where this has been quantified.

F. It is confusing that A-92 increases p-T899 GCN2 levels in Fig. 10A. Additionally, why were p-S51 levels not probed in 10A?"

Response. We have performed replicate experiments and quantified these blots and included eIF2 α blots. It appears that A-92 is not causing phosphorylation of GCN2, which is consistent with our data where we used it to antagonise RAFi-induced GCN2 activation. GCN2iB does cause a small but significant increase in p-GCN2 levels and this is amplified down the pathway by increases in ATF4 expression.

Minor:

-Figure 7. What is +ve ATP in the first lane?

Response. This is a positive control for activation of GCN2 in the *in vitro* assays. A high dose of ATP is sufficient to activate this enzyme.

Reviewer #2 (Remarks to the Author)

Although targeting RAF kinase, particularly BRAF(V600E) mutant with RAFi for treating cancers has achieved promising outcome, drug resistance and toxicity remains a major challenge in current clinic practice. Understanding underlying mechanisms has important implications for both cancer research and treatment. In this manuscript entitled “RAF inhibitors activate the integrated stress response by direct activation of GCN2”, Gilley et al. found that RAFi induced ERK-independent/eIF2 α -dependent expression of ATF4 and CHOP through activating unfolded protein response. Further, they showed that GCN2 was a key factor for RAFi-induced ATF4 and CHOP expression by using GCN2 inhibitor, GCN2 RNAi, and GCN2 KO system. More importantly, using biochemical, molecular modelling and mutagenesis approaches, they demonstrated that RAFi directly associated with GCN2 and enhanced GCN2 dimerization, which leads to the paradoxical activation of GCN2 as RAFi does on RAF kinase. Overall, this is a very interesting study with solid biochemical data, and somehow contributes targeted cancer therapy with RAF inhibitors, since RAFi-induced ISR (integrated stress response) may provide cancer cells a time window for developing acquired resistance. Although it has been reported by other groups that RAFi activates GCN2-eIF2 α -ATF4 axis and hence induces ISR (i.e. Nagasawa et al., BBRC, 2017; Nagasawa et al., iScience, 2020), it is novel that authors identified GCN2 kinase as the direct target of RAFi and GCN2 was transactivated by RAFi. However, there are some significant issues needed to be addressed in this manuscript before considering for publishing.

Response. We thank the reviewer for their kind comments and recognising the novelty of our work in showing ‘GCN2 kinase as the direct target of RAFi and GCN2 was transactivated by RAFi’.

Major points:

1. The most important finding of this study is that RAFi binds to and transactivates GCN2 through inducing dimerization as it does on RAF. However, authors’ data support this finding is not solid. To strengthen this point, firstly they need to provide the data of RAFi affinity with GCN2, since they already purified GCN2 protein and could get this data easily by using Surface Plasmon Resonance (SPR) or Microscale Thermophoresis (MST) approaches.

Response. First of all, and to be clear, we do not have any evidence to support RAFi inducing dimerization of GCN2 as is proposed for RAFi promoting RAF dimerisation. Human GCN2 is an obligate dimer and our in vitro kinase assays were performed with purified dimeric GCN2.

We envisage that at low doses RAFi bind to the ATP binding site of one protomer in a pre-existing GCN2 dimer and inhibit it but also promote allosteric changes in the drug-free dimer partner, resulting in its activation, presumably through alignment of the hydrophobic spines and repositioning of helix α C.

Evidence supporting this includes:

(i) the ability of A-92, an ATP competitive GCN2 inhibitor to antagonise RAFi-induced GCN2-ISR activation;

(ii) our molecular modelling is consistent with Dabrafenib, LY3009120 or GCN2iB binding and promoting the assembly of the R-spine whereas GDC-0879 is not predicted to assemble the R-spine and does not activate GCN2 in vitro or in cells. Similarly, A-92 is

not predicted to assemble the R-spine and does not activate GCN2 in cells but rather inhibits it;

(iii) The detailed SPR and MST assays proposed are beyond the scope of our cell-based study. However, the parallel paper (<https://doi.org/10.1101/2024.08.14.606984>) from our collaborator Glenn Masson focuses on biophysical approaches to provide critical further molecular and structural insights. First, Thermal Shift Assays show that Dabrafenib, Encorafenib and GCN2iB all produce significant shifts in stability of purified full length GCN2 dimers, indicative of binding, while GDC-0879 produced no shift. Second, HDX-MS analysis shows that Dabrafenib, Encorafenib and GCN2iB all bind within the kinase of domain of GCN2, causing similar large reductions in solvent exchange in peptides surrounding the ATP-binding pocket (residues 587-592 (P-Loop), 803-818 (Activation Loop), 836-876 (DFG Loop), 902-914 (T904 auto-phosphorylation site).

Whether RAFi-induced GCN2 phosphorylation is “cis-autophosphorylation” where one GCN2 kinase phosphorylates its own T899 residue or “trans-autophosphorylation” with the T899 residue of the dimer partner being phosphorylated remains to be resolved. T899 phosphorylation of the drug-bound dimer partner by the drug-free allosterically activated partner seems more plausible to us. A variation on this model is presented in the companion paper by Glenn Masson (<https://doi.org/10.1101/2024.08.14.606984>) in which he demonstrates that in the test tube at least RAFi binding to one GCN2 dimer may make it a better substrate for phosphorylation by another GCN2 dimer. Whether dimer-to-dimer phosphorylation occurs in cells remains to be seen.

Regardless of these variations on the model, subsequent dissociation of RAFi (by competition with ATP) from ‘poised, phosphorylated’ GCN2 would then allow it to phosphorylate eIF2a. This is consistent with the new *in vitro* GCN2 kinase data (Figure 7C); in our previous very little RAFi-driven phosphorylation of eIF2a was observed at 10 μ M ATP, though this supported robust GCN2 T899 phosphorylation. P-eIF2 α was only observed at 100 μ M ATP, consistent with ATP competing RAFi off of phosphorylated GCN2, essentially removing inhibition and allowing it to phosphorylate its substrate. We now describe these various scenarios in the discussion.

Secondly, for their molecular modelling of RAFi-GCN2 complex, it should be validated by using mutagenesis approach since it’s just a prediction that needs to confirm by using other approach.

Response. We agree with the referee that our models must be validated by using mutagenesis. The modelling in Figure 8 confirms our cellular studies since those inhibitors that activate GCN2 in cells (Dabrafenib, LY3009120 or GCN2iB) bind in a manner that predicts assembly of the R-spine whilst those that don’t (GDC-0879 and A-92) do not. The models in Figure 8 make certain predictions that we have validated by mutagenesis in Figure 9. First they, they predict that a kinase dead mutant of GCN2 will fail to restore RAFi-induced GCN2 activation; this is confirmed in Fig 9A. Second they predict that mutating the GCN2 gatekeeper residue should modulate sensitivity to RAFi. This turns out to be correct (Fig 9B) since a smaller Ala residue (GCN2 M802A) is sensitised (left-shifted) to Dabrafenib whereas the bulkier GCN2 802F is constitutively active (Fig 9A).

In addition, the RAFi-driven transactivation of GCN2 is also needed to confirm by mutating the GCN2 dimer interface.

Response. GCN2 is an obligate dimer. Our collaborator Glenn Masson, a world expert on GCN2 structural biology, has found that mutations that disrupt GCN2 dimerisation destabilise the protein so we haven't been able to do these experiments. This was highlighted in the original Discussion.

2. Some experiments (i.e. experiments in Fig1~4) were carried out on single cell line, NCI-H358. To exclude the cell line-specific effect, those data should be validated by using at least one more cell line. Since RAF inhibitors have been applied to treat BRAF(V600E)-mutated melanoma and other cancers, we suggest, authors may use a BRAF(V600E)-harboring melanoma cell line to validate those data, which could also elevate the clinic significance of their findings.

Response. Here we have shown that 8 of 10 RAFi tested activate GCN2 and the ISR, including all three clinically approved RAFi and some paradox breakers; furthermore this occurs at clinically relevant doses. We have demonstrated using pharmacological and molecular genetic interventions that GCN2 is required for activation of the ISR by RAFi and shown that RAFi bind to the ATP-binding site of GCN2 to directly activate the enzyme. We have supported this with mechanistic models that complement our analysis and have prompted further mutational studies of the gatekeeper residue which have supported our model. We feel that this body of work is in itself a significant advance.

Our results showing that RAFi activate GCN2 and the ISR includes studies in NCI-H358 (NSCLC KRASmut), HCT116 (colorectal cancer or CRC, KRAS mut) and HT29 (CRC BRAFV600E) as well as immortalised MEFs. Since BRAF inhibitors are now approved for BRAFV600E-harboring colorectal cancer we tested RAFi in clinically relevant cells lines for BRAFi treatment.

It is likely that different tumours or even sub-types of the same tumour will respond differently to paradoxical activation of GCN2 by RAFi, depending on a variety of factors such as whether key cell cycle checkpoint machinery is intact. Whether tumour cells respond to RAFi by undergoing growth arrest as was seen in NCI-H358 or not, is not a matter that will be resolved by validating one more cell line. It will require analysis of panels of tumour cells lines with defined oncogene or tumour suppressor mutations and comparisons with isogenic derivatives. For example, even testing the antiproliferative effects of BRAFi in BRAF-mutant melanoma or CRC will first need to take account of the potent anti-proliferative effects of 'on-target' inhibition of BRAFV600E by comparing WT and GCN2 null derivatives. We are starting this work now, but it clearly falls beyond the scope of this current paper.

Minor points:

1. Line 55-57, "BRAFV600E/K.....signal as constitutively active monomers." The monomer hypothesis of BRAF(V600E) was raised up for explaining the RAFi resistance before, but so far there's no solid data supporting this opinion. On the other sides, several groups had provided strong evidence that RAF kinases as well as their mutants including BRAF(V600E) function as dimers or oligomers to activate MEK (Diedrich et al., EMBO J, 2017; Yuan et al., oncogene, 2018; Cope et al., Chembiochem, 2019; Lauinger et al., Sci Adv, 2023). This should be corrected.

Response. – ". The reviewer makes an important point. However, the manuscript is not focused on RAF oligomers, dimers or monomers but rather focuses on the ability of RAFi to activate GCN2. The reviewer is correct that these BRAF mutants can form dimers or oligomers. This sentence refers to the clinical success of RAFi which is in tumours with BRAFV600E/K where the drugs completely shut down ERK signalling, suggesting their success is due to targeting the monomer.

In the interests of space and focus on GCN2 we have corrected the text to indicate that "BRAFV600E/K..... **can** signal as constitutively active monomers".

2. Figure 5c, the quality of phospho-GCN2 blot is low, please repeat this blot and get a better image. Also, samples for WT, GCN2 KO should be run on one blot, otherwise, they are not comparable. This issue also existed in 5D.

Response.

We have generated new data for Dabrafenib in Figure 5B in which WT and GCN2 KO cell samples are resolved on the same gel and we have quantified 3 independent experiments to show the increase in p-GCN2 (T899) is significant. This is also repeated in Figure 5C with a panel of RAFi and controls and again has been quantified to show the increases in p-GCN2 (T899), p-eIF2 α , ATF4 and CHOP are all statistically significant.

3. In Figure 5c-d, Figure 6b-c, Figure 9a-c, and Figure 10a, is HSP90 used a loading control or ISR marker?

Response. A loading control

4. Figure 7c, it's better to re-organize the sample orders, samples treated with a same drug at different concentrations should be put together, which will be more convenient for comparing drugs' effect.

Response. We respectfully disagree. The best way to compare differences in potency of different drugs is to have the four drugs grouped together for each dose as we have done. In this figure we wanted to compare the different drugs side by side and highlight that at 1 μ M Dab, Enco and LY, doses at which we see maximal GCN2 activation in cells, we also see maximal activation *in vitro*. This side-by-side loading also allows a clear comparison with GDC-0879, which does not activate GCN2 compared to the other RAFi and fails to activate GCN2 in cells. We note that the other two referees were content with this loading order.

5. Figure 9a, the GFP-GCN2 blot needs to be improved.

Response. We have repeated this experiment using an alternative p-GCN2 T899 antibody (CST) and have used the total GCN2 antibody to show total protein levels as it appears cleaner than the GFP antibody.

6. Figure 10a, it's better to run all samples on one blot.

Response. It is important to recognise that quantifying and comparing different experiments inevitably means that data is being pooled from multiple experiments resolved on different gels. But, as requested, we have done this for Dabrafenib and made a new figure (Fig 5B) and this has been quantified (together with other experiments) to show the increases in p-GCN2 (T899), p-eIF2 α and ATF4 are all statistically significant. For other experiments there were too many samples to run on one gel; in these cases we

have always exposed all gels at the same time so the samples are directly comparable and used these for quantification from multiple experiments. In addition, even samples compared between two gels have been quantified and presented as a ratio to relevant loading controls.

7. Reference#39, it's a repetitive citation. Please delete it.

Response. Thank you, we have done this.

Reviewer #3 (Remarks to the Author)

The manuscript by Gilley et al. describes studies of RAF inhibitors prompted by observations of reduced DNA replication that the authors use to zero in on an unexpected activation of the ISR. Using genetic and biochemical approaches the authors show that interactions with GCN2 are important to activate the ISR and block DNA replication in the RAFi treated cells.

Overall, the work was thorough and methodical and I had very few concerns. Below are some queries and suggestions.

Response. We are grateful to the referee for their positive comments and recognising our novel study is thorough.

1. In Figure 2, BiP seems to be induced by Selumetinib after 24 hours in panel A, but it is not induced in panel B. Why?

Response. Looking again at our replicate experiments Figure 2B is not representative of what we see with BiP; the small affect of Sel on BiP in Fig 2A is an outlier so we have removed the BiP panel from Figure 2A to avoid any confusion. The main point of Figure 2A is to show the strong Dabrafenib-induced, Sel-independent expression of ATF4 in the replicate samples that were used for the RNAseq (Supp figure 3).

2. In general the authors rely on western blots where the differences are usually quite stark. However, in some instances that are more subtle quantification would be helpful.

Response We have quantified a range of relevant blots in response to one other referee's comments to show that subtle differences are indeed statistically significant. This is especially the case for p-S51 eIF2 α which is notoriously difficult to quantify as reviewed by Bertolotti et al https://doi.org/10.1007/978-1-0716-1975-9_1 This comprehensive methods chapter states "phosphorylation of eIF2 α remains difficult to detect and quantify, because of its transient nature and because sub-stoichiometric amounts of this modification are sufficient to profoundly reshape cellular physiology". This review also presents data showing a 2-fold increase in p-S51 eIF2 α with Tunicamycin (an ER stressor and PERK activator) as being a 'typical' response. Indeed, in our experiments Tunicamycin did indeed drive a 1.8 to 2-fold increase in p-S51 eIF2 α but this 'modest' signal is amplified down the pathway resulting in substantial increases in both ATF4 and

CHOP expression. Thus, our results with RAFi are significant and benchmarked against a recognised standard in the field.

3. In Figure 5, the authors claim that GCN2iB causes an increase in ATF4 and CHOP. While the increase in ATF4 is clear enough, I don't see a significant increase in CHOP. In other figures this induction is time sensitive and lags after ATF4 so perhaps it is a timing issue. Regardless, can the authors provide stronger support for this statement? **Response.** The referee is quite correct. The induction of CHOP with selective GCN2 activators is much weaker than with ER stressors, perhaps reflecting input from other arms of the UPR (ATF6 or IRE1). We have changed the text to reflect this.

4. There are a large number of inhibitors and signalling outputs tested which can be somewhat confusing. I suggest the authors draw a model figure at the end summing up the major findings of the work. Highlighting, in particular, the potential clinical impacts of their work.

Response. This is an excellent suggestion and we have now included such a figure (Fig 11).